# Permafrost Deformation Monitoring Along the Qinghai-Tibet Plateau Engineering Corridor Using InSAR Observations with Multi-Sensor SAR Datasets from 1997–2018

**DOI:** 10.3390/s19235306

**Published:** 2019-12-02

**Authors:** Zhengjia Zhang, Mengmeng Wang, Zhijie Wu, Xiuguo Liu

**Affiliations:** 1School of Geography and Information Engineering, China University of Geosciences, 388 Lumo Road, Wuhan 430074, China; zhangzj@cug.edu.cn (Z.Z.); wangmm@cug.edu.cn (M.W.); 2Artificial Intelligence School, Wuchang University of Technology, No. 16 Jiangxia Avenue, Wuhan 430223, China; 3College of Resources Engineering, Longyan University, Longyan 264012, China

**Keywords:** InSAR, Qinghai-Tibet Engineering Corridor, deformation, permafrost

## Abstract

As the highest elevation permafrost region in the world, the Qinghai-Tibet Plateau (QTP) permafrost is quickly degrading due to global warming, climate change and human activities. The Qinghai-Tibet Engineering Corridor (QTEC), located in the QTP tundra, is of growing interest due to the increased infrastructure development in the remote QTP area. The ground, including the embankment of permafrost engineering, is prone to instability, primarily due to the seasonal freezing and thawing cycles and increase in human activities. In this study, we used ERS-1 (1997–1999), ENVISAT (2004–2010) and Sentinel-1A (2015–2018) images to assess the ground deformation along QTEC using time-series InSAR. We present a piecewise deformation model including periodic deformation related to seasonal components and interannual linear subsidence trends was presented. Analysis of the ERS-1 result show ground deformation along QTEC ranged from −5 to +5 mm/year during the 1997–1999 observation period. For the ENVISAT and Sentinel-1A results, the estimated deformation rate ranged from −20 to +10 mm/year. Throughout the whole observation period, most of the QTEC appeared to be stable. Significant ground deformation was detected in three sections of the corridor in the Sentinel-1A results. An analysis of the distribution of the thaw slumping region in the Tuotuohe area reveals that ground deformation was associated with the development of thaw slumps in one of the three sections. This research indicates that the InSAR technique could be crucial for monitoring the ground deformation along QTEC.

## 1. Introduction

Permafrost, defined as soil or rock ground that remains frozen (ground temperature below 0 °C) for two or more consecutive years [1,2], has the potential to affect the global climate [3,4], carbon balance [5], and water-heat balance [6]. The Qinghai-Tibet Plateau (QTP) has the largest extent of permafrost outside the polar regions, with 50% of the QTP’s area underlain by permafrost. With the implementation of western development strategy and the One Belt and One Road strategy, several key engineering projects have been conducted on this fragile and harsh environmental plateau, such as the Qinghai-Tibet Railway (QTR) [7,8], the Qinghai-Tibet Highway (QTH) [9], oil pipelines [10] and electric transmission lines [11]. Along the QTR from the Chumaerhe to Fenghuo Mountain is the significant section of Qinghai-Tibet Engineering corridor (QTEC) [12,13]. In recent years, with the global warming, the increase of human activities, and the operation of permafrost engineering, the permafrost has become seriously degraded, intensifying the permafrost engineering instability, land desertification and soil moisture loss [14]. Therefore, long-term permafrost measurement along the QTEC is of great importance for permafrost environment protection, climate change and cold-region hazard prevention.

Traditional geodetic measurement methods such as levelling and the global position system (GPS) surveys, can achieve high-precision monitoring. However, these point-based geodetic measured methods are limited to discrete points on fixed routes and are time consuming. Compared with those methods, the satellite remote sensing provides a valuable tool for observing large and hard-to-access areas with high spatial and temporal resolution [15]. Synthetic aperture radar interferometry (InSAR) is a promising technique that can be used to monitor slow ground deformation with millimeter accuracy by analyzing the phase information from two SAR images [16]. Due to the advantages of large coverage, high resolution and measurement accuracy, InSAR has been used to measure surface deformation over larger areas induced by earthquake [17], volcanoes [18], and land subsidence [19,20]. It has also been adopted to determine the ground deformation in permafrost regions [15,21,22].

To mitigate the intrinsic limitations of the traditional differential InSAR (DInSAR) (spatial-temporal decorrelations and atmospheric delay) [23], time-series InSAR techniques such as persistent scatterer interferometry (PSI) [24,25], the small baseline subset interferometry (SBAS) [26,27], multi-temporal InSAR (MTInSAR) [28], have been proposed by analyzing the time series interferometric phase on stable objects, such as buildings, rocks and roads.

Due to the merits of time-series InSAR, many studies have used it to retrieve surface deformation information related to permafrost thawing and freezing in QTP [29,30,31,32,33,34,35,36,37,38,39] (Table 1) and other permafrost regions [21,22,40].

The studies listed in Table 1 preliminarily explored the deformation of the permafrost region using time-series InSAR. Unfortunately, the most of the above-mentioned literatures have only focused on permafrost deformation monitoring in QTP over a short period of time such as from 2004 to 2009 with ENVISAT images or from 2007–2010 with ALOS-1 images, or from 2014 to 2016 with TerraSAR-X images. Long-term Permafrost thaw deformation on the QTP and the relationship between permafrost deformation and QTP engineering are still poorly quantified and understood. It is necessary to focus on the latest development and the temporal evolution of ground deformation of the permafrost region in QTP. With the launch of new SAR satellites such as Sentinel-1A/B [41], more SAR images with short repeat cycles (six days) can be obtained, which are suitable for determining the ground deformation in permafrost regions. Daout et al., developed a method to enhance InSAR performances for such difficult terrain conditions and construct an 8 year timeline of the surface deformation over a 60,000 km^2^ area [39]. Rouyet et al. used the InSAR to investigate the seasonal ground deformation in and around Adventdalen with TerraSAR-X StripMap Mode (2009–2017) and Sentinel-1 Interferometric Wide Swath Mode (2015–2017) SAR images [15]. Combining the archived SAR images, long-term ground deformation in the permafrost region can be determined.

The objectives of this paper were to retrieve the surface deformation along QTEC from the Wudaoliang to the Tuotuohe section over a 20-years period using time-series InSAR technique and to analyze temporal evolution of the QTEC deformation. More than 90 SAR images, including ERS-1, ENVISAT, and Sentinel-1A, were collected to jointly retrieve the feature of ground deformation from 1997 to 2018. A hybrid time-series methodology taking advantage of the merits of PSI and SBAS was used to identify more measurement points [42]. Moreover, a piecewise deformation model combining a seasonal deformation term related to active layer thawing and freezing and linear subsidence component related to permafrost thawing is introduced. The spatiotemporal feature of the ground deformation along the QTEC and its relationships with permafrost engineering and permafrost distribution were analyzed.

## 2. Study Area and Datasets

### 2.1. Study Area

The permafrost region along the Wudaoliang-Tuotuohe section of QTEC was chosen as the study area. The area is in the in the Hoh Xil mountain area between the Kunlun Mountain and Tanggula Mountain ranges and is the source area of the Yangtze river, which is in the northern part of the QTP [12]. The QTR is a high-elevation railway connecting Xining to Lhasa, with the length of 1956 km. About 550 km length of the QTR is laid on the discontinuous permafrost [7]. The QTR from Wudaoliang to Tuotuohe section began to construct in 2001 and completed in 2006. Figure 1 provide a topographic map of the study area, with an average elevation of more than 4500 m above the sea level. Underground ice developed extensively in this region [5]. Several thermokarst lakes have developed, such as Zuonai Lake, Kusai Lake and Salt Lake (Figure 1). The active layer thickness (ALT) varies from 0.8 to 4 m with a mean of about 2 m [43]. The typical ground features in our study area can be classified into six landcover types: alpine meadow, alpine desert, Thermokarst Lake, QTR, QTH, and electric transmission power line (Figure 2). This area is dominated by sub frigid semi-arid climate with the mean temperature of about −3.8 °C [44]. The annual mean precipitation varies from 50 mm to 400 mm, concentrated in the summer season [36]. The amplitude of Sentinel-1A, shuttle radar topography mission (SRTM) digital elevation map (DEM) data [45] and the slope of the QTEC are show in the bottom of Figure 1. In the QTEC, several permafrost engineering structures have been constructed that have considerably influenced the stability of the permafrost. In the SAR images, the QTR is a bright line as shown in Figure 1. The other engineering structures cannot be easily observed in medium resolution SAR images.

Within the QTEC coverage, several key developmental projects have been constructed, such as QTR, QTH, and electric transmission power line (Figure 2d–f). Due to the constructions of those permafrost engineering structures, the original hydrothermal balance of permafrost has been destroyed and the permafrost has begun to degrade. Studies have showed that the ground deformation rate of the permafrost along QTR can reach −10 mm/year in some sections [46]. The study area is an overlap of the available SAR images. About 110 km of the QTEC region from Wudaoliang to Tuotuohe was selected as the study object. The daily air temperature in Wudaoliang weather station from 1997 to 2018 was collected. Figure 3 shows the daily air temperature in our study area from 1997 to 2018.

### 2.2. Datasets

To reveal the ground deformation in the study area in the selected 20-year period, SAR images from three different satellites were collected. There are ERS-1 SAR images acquired from October 1997 to December 1999; ENVISAT SAR images from November 2004 to July 2010, and Sentinel-1A SAR images from April 2015 to December 2018. The coverages of the above SAR stacks are shown in Figure 1. The amplitude of the Sentinel-1A along QTEC is show in the bottom of Figure 1. The QTR and QTH can been easily observed due to their strong back scattering. The acquisition parameters of the three SAR images are listed in Table 2. Unfortunately, time gaps, where no SAR images are acquired, exist 2002–2004 and 2009–2015. SRTM DEM data with a spatial resolution of 30 m were adopted to remove the topographic phase.

## 3. Methodology

### 3.1. InSAR Processing

Studies have demonstrated that the main challenges and limitations of the InSAR technique in detecting the ground deformation in the permafrost region are the serious temporal decorrelation and non-linear deformation trends caused by the seasonal thaw-freeze process of active layer [31,37,46,47]. It is difficult to obtain sufficiently stable measurement points due to the above limitations. In this study, the small baseline strategy was applied to suppress the temporal decorrelation.

Firstly, all the SAR images were co-registered. Then, a multi-temporal InSAR data processing strategy was used to retrieve ground deformation. Considering the different attribute of SAR stacks with different wave lengths, different small baseline strategies were adopted for those SAR stacks [48]. Through previous studies, the temporal decorrelation is serious in permafrost region, so the temporal baseline (350 days) is no longer than one year. Consideration the orbit accuracy of different sensors and the time sampling of SAR images, the normal baseline threshold values are 800 m, 500 m, and 200 m for ERS, ENVISAT and Sentinel-1A, respectively. For ERS-1 and ENVISAT, the multi-looking with 5 × 1 looks in the azimuth and range direction was performed, respectively. For Sentinel-1A, the multi-looking with 1 × 4 looks in the azimuth and range direction were performed. After all the interferograms have been generated, each of the interferograms were checked, and the interferograms with serious temporal decorrelations were rejected for deformation retrieval. Finally, we obtained a total number of 17 ERS-1 interferograms (normal baseline < 800 m and temporal baseline < 350 days), 105 ENVISAT ASAR interferograms (normal baseline < 500 m and temporal baseline < 350 days), and 131 Sentinel-1A interferograms (normal baseline < 200 m and temporal baseline < 350 days). Figure 4 shows the spatial and temporal baseline configuration of the three SAR stacks. The differential interferometric phase is generated by removing the topographic phase from the interferograms using the SRTM DEM data. To suppress the noise in the interferograms, the Goldstein filtering method was applied [49].

### 3.2. Seasonal and Long-Term Deformation Model

The thaw-freeze process of the active layer is complex and correlated with many factors, such as vegetation, snow, soil moisture, soil properties and temperature [7]. In permafrost regions, the seasonal deformation component is larger than the annual deformation. Therefore, using an appropriate seasonal phase model to monitor the thawing-freezing process of the permafrost is essential. Mathematical models, such as the sinusoidal model [33,36,50] and cubic term model [31] have been proposed to retrieved the seasonal deformation of permafrost. However, the seasonal deformation term is much complicated and is closely related to Environmental and climatic factors, such as temperature, soil moisture. These environmental and climatic factors should be considered. Liu et al. [51] introduced a seasonal model based on the Stefan model in the Alaska permafrost region, which describes the relationship between the thaw depth and the square root of the accumulated degree days of thawing (ADDT). The Stefan equation is widely used to estimate the thaw depth. This deformation model is based on the cumulative temperature and is reasonable, which have been successfully applied in QTP regions [37,38]. In this study, we adopted a deformation model combining a linear subsidence term for the long-term permafrost thaw subsidence and seasonal deformation term for the seasonal thawing and freezing of the active layer.

The deformation model is defined as follows:(1)$$d_{s}=R\cdot t+A_{t}\cdot \sqrt{\mathrm{ADDT}(t_{1})}-A_{f}\cdot \sqrt{\mathrm{ADDF}(t_{2})}+\varepsilon$$
where, *R* is the long-term deformation rate, *A_t_* and *A_f_* are the thawing and freezing deformation coefficients, respectively; and ADDT and ADDF are the accumulated degree days of thawing and freezing, respectively. ADDT reaches its the maximum at the end of the thawing season. The daily ADDT and ADDF were calculated based on the air temperature measured at the Wudaoliang Meteorological station. Due to the sporadic acquisitions of ERS-1 images, a deformation model without a seasonal term was used for ERS-1 datasets.

### 3.3. Calculation of ADDT and ADDF

The thawing and freezing onsets of the active layer are fixed as 1 May and 15 September, respectively [21,37]. However, the freezing and thawing onsets change every year in the QTP. Error would occur if we assume that the thawing and freezing onsets were the same in every year. In QTP, the length of freezing season is longer than the of thawing season. Generally, a uniform thawing and freezing onsets of the active layer are chosen based on temperature observation data. In this study, we first used the following model to monitor temperature:(2)$$T\left(t\right)=a_{0}+a_{1}\cos\left(t\cdot w\right)+a_{2}\sin\left(t\cdot w\right)$$
where, *T* (*t*) is the temperature on day *t*. $a_{0},\;a_{1},a_{03},\;and\;w$ are the parameters. For each year, we used this model to monitor the annual temperature and identify the thawing and freezing onsets in each year.

Figure 5 shows the time-series temperature of each year from 1997 to 2018. Most of the 20 years of temperature data were modeled accuracy with a coefficient of determination (R^2^) > 0.9 and root mean square error (RMSE) < 2.7. Figure 5 shows that the onsets of thawing and freezing changed every year. Through the monitoring results, we identified the onsets of thawing and freezing and relatively accurately calculate the ADDT and ADDF each year.

### 3.4. Time-Series InSAR Method

#### 3.4.1. Coherence Point (CP) Selection

CPs are those points with high coherence and stable amplitude value during the whole observation period. In permafrost area, the ground feature includes four types: permafrost engineering, Thermokarst Lake, alpine meadow, alpine desert (Figure 2). In order to exclude the water bodies, vegetated areas and other decorrelated areas from the CPs, the thresholds of coherence and the dispersion of amplitude are both used to identify the CPs. In this paper, the coherence threshold value is 0.65 and the dispersion amplitude threshold value is 0.25.

#### 3.4.2. Topographic and Orbit Error Removal

The atmospheric delay is influential in high latitude mountainous regions. Our study area has an average elevation of over 4400 m with some mountains. In the mountainous areas, the stratified troposphere can produce serious atmospheric delays in the interferograms. Obvious residual orbital phase was in some interferograms. In this paper, to remove those phase ramps, we applied a phase ramps correction model combining a biquadratic model for orbital phase ramps and a linear model for elevation dependents errors [52]:(3)$${\varphi (x,y)}_{ramp}=a_{0}+a_{1}\cdot x+a_{2}\cdot x^{2}+a_{3}\cdot x\cdot y+a_{4}\cdot y+a_{5}\cdot y^{2}+a_{6}\cdot h+\varepsilon (x,y)$$
where, $\varphi {\left(x,y\right)}_{ramp}$ is the modeled phase ramps, *ε*(*x, y*) is the random phase error, *a*_i_ represents the estimated parameters. The interferograms with obvious phase ramps were corrected using this model. After that correction process, we assumed that most of the topography related phase errors (DEM error and atmospheric delay) have been removed.

#### 3.4.3. Atmospheric Phase Screen (APS) Removal

The residual phases for each interferogram were calculated by subtracting the estimated LP deformation and topographic error phase from the differential interferograms and unwrapped by the sparse Minimum Cost Flow (MCF) method [53]. The atmospheric phase was considered to consist of two components: topography related and non-topographic related [54]. The two components were estimated separately. The topography related component can be estimated by the M-estimated. The non-topographic related atmospheric phase component is highly correlated in space but poorly in time, which can be estimated from the resultant phase based on the low pass filtering operation in spatial domain and high pass filtering operation in the temporal domain. After removing the APS from each interferogram and applying additional least-square estimation, we obtained the time-series deformation map.

#### 3.4.4. Parameter Estimation

After identification of the CPs, all the CPs were connected to further remove the effects of the atmospheric delay. The differences of those differential interferometric phase between the neighboring CPs in the ith interferograms can be written as:(4)$$\Delta \varphi^{i}{}_{model}=\Delta \varphi_{def,i}(\Delta R,\Delta A)+\Delta \varphi_{topo,i}(\Delta \tau )+\Delta \varepsilon_{i}$$
where, $\Delta \varphi^{i}{}_{model}$ is the model phase difference of the neighboring two CPs in the ith interferograms. ΔR and ΔA are the differential rate of linear deformation and seasonal deformation (Equation (1)), respectively; $\Delta \varphi_{topo,i}$ is the residual topographic phase due to the DEM error (Δτ); $\Delta \varepsilon_{i}$ denotes the phase noise.

The identified CPs were firstly connected based on the Delaunay triangulation network. Then, the differential phase of all the edges were calculated, which is beneficial to further remove atmospheric and orbital errors. The parameters ΔR,ΔA, and Δτ were optimally estimated for all of the edges using the periodogram approach [24]. After the differential parameters of all the edges had been estimated, a quality test was performed to reject links with temporal coherence lower than the threshold. In our experiment, the temporal coherence the threshold value is 0.7. Moreover, the edges with the length larger than 3 km were also rejected to mitigate the spatially-correlated phase errors, such as atmospheric delay. Then, a reference point was selected and we used the least-squares estimation to derive the parameters (*R*, *A*, and τ) of each point. We applied the temporal coherence as a weighting function during the inversion process. The estimated *R* and *A* are along the slant light of sight (LOS) direction. We assumed that the detected deformation is in vertical direction, and the LOS estimated deformation was converted to the vertical direction by dividing the cosine of the average incidence angles. The specific procedures of the approach are illustrated in Figure 6.

## 4. Results and Analysis

### 4.1. InSAR Results

Using the time-series InSAR method described above, the estimated average ground deformation rate along QTEC from the Wudaoliang to Tuotuohe sections using three C-band SAR stacks from 1997 to 2018 have been generated, including the deformation rate from 1997 to 1999 calculated with ERS-1 data (see Figure 7a), the deformation rate from 2004 to 2010 calculated using ENVISAT data (see Figure 7b), and the deformation rate from 2015 to 2018 calculated using Sentinel-1A data (see Figure 7c). The reference point (red star, Figure 7) was selected at the railway bridge. Negative deformation velocity represents an increasing distance with time away from the radar satellite; and positive deformation velocity indicates a decreasing distance towards the radar satellite. About over 100 km of the QTEC have been monitored. 40,760 CPs were detected for the ERS-1 along QTEC, and 125,522 CPs were detected for the ENVISAT, 217,096 CPs were selected for the Sentinel-1A. Figure 7d–f depict the estimated DEM errors of ERS-1, ENVISAT, and Sentinel-1A, respectively. The estimated DEM error ranged from −20 m to 10 m in most of the study area, which is consistent with the relative accuracy of the SRTM DEM. Most of the CPs are corresponded to QTR and QTH embankments, rocky mountains, and other artificial engineering structures. Before the 1999, the QTR and QTH were not completely constructed and few CPs were detected for ERS-1 data. For the Sentinel-1A, more SAR images are collected per year and more interferograms with less baseline were generated, so more CPs were detected.

The ground deformation rate along the QTEC ranges from −10 to +10 mm/year during the 1997–1999 observation period derived from ERS-1 data. For the ENVISAT and Sentinel-1A experiments, the estimated deformation velocity was primarily in the range of −20 to +10 mm/year. The spatial distribution of the deformation before 2004 was quite different from those after 2004, and the deformation rate of the ERS-1 was inaccurate due to the few SAR datasets and heterogeneous spatial-temporal baseline.

In this study, we choose the QTR as an example to analyze the deformation of permafrost engineering. Figure 8 shows the deformation rate profile of QTR (from points P1 to P1 in Figure 7c) from 1997 to 2018. Through the above result, we found that before the opening of the QTR in 2006, the ground deformation along was relatively minimal. After the opening of the QTR, the overall mean deformation rate at the beginning and the end of QTR was within 10 mm/year. Four regions with obvious ground deformation in recent year have been detected. Regions A (Beiluhe) and B (south of Fenghuo Mountain) showed an obvious subsidence area, with the largest deformation rate being 15 mm/year. From 2015 to 2018, two more QTR section with ground deformation, Region C (Tuotuohe) and D, were detected, with the maximum ground displacement velocity over 17 mm/year. In some sections of the QTR, some cracks were found on the embankment shoulder and slopes through our field investigation. Long term monitoring is necessary in those areas. The surface subsidence along the embankment of QTR was primarily in the range of −20 mm/year to 5 mm/year. Human activities, such as embankment construction and railway operation, disrupt the original hydrothermal balance of the active layer, contributing to the obvious ground settlements [31].

### 4.2. Regional Analysis

Obvious deformation along QTEC was detected in three areas as enclosed by the red dashed ellipses in Figure 7 corresponding to the regions prone to subsidence based on previous investigations [37,46], i.e., Beiluhe, Fenghuo Mountain, and Tuotuohe areas. To analyze the deformation pattern along the QTEC, the detected obvious ground deformation regions in Beiluhe (Figure 9 and Figure 10), Fenghuo Mountain (Figure 11), and Tuotuohe (Figure 12) are analyzed in detailed. A closer analysis of those three areas is provided below.

#### 4.2.1. Beiluhe

The Beiluhe basin region is in the tundra of Hoh Xil and is underlain by cold permafrost. The terrain is relatively flat and most of the slope is less than 40°. The soil moisture content in the surface is high in the summer season and can reach 0.3 [55]. The vegetation coverage ranges from 0.3 to 0.9, which would contribute to serious temporal decorrelation. The Beiluhe permafrost region has been undergoing serious ground deformation in recent decades [5,7].

Figure 9 shows the mean LOS displacement rate in the Beiluhe permafrost region. Most of the selected points were located on the embankment of QTR and QTH. Fewer CPs are located on the alpine meadow areas due to serious temporal decorrelation. The primary displacement rate was in the range of −6 to 5 mm/year during 1997–1999 from ERS-1 dataset. The ENVISAT and Sentinel-1A results showed obvious ground deformation trend, with the larger deformation rate of −10 mm/year and −15mm/year respectively. Most of the deformation points are in the south of the region, which is consistent with the finding reported in previous studies [37,43].

During the field investigations in 2014 and 2015, some surface cracks or fissures of about 20 cm along the QTEC and alpine meadow regions were found in the Beiluhe regions, as shown in Figure 10a–c. The long-term active layer thawing-freezing effect caused long cracks in the alpine meadow areas.

#### 4.2.2. Southern of Fenghuo Mountain

The Fenghuo Mountain, with an average elevation of more than 5000 m, is to the southeast of Hoh Xil, 380 km away from the city of Golmud. The 1.33 km long QTR Fenghuo Mountain tunnel was successfully traversed on 19 October, 2002. Figure 7 shows that the ground along the QTR Fenghuo Mountain tunnel was stable from 2004 to 2018 and no obvious deformation trend has detected. In the south of Fenghuo Mountain, visible ground displacement was found per the InSAR results. Figure 11 shows the mean LOS displacement rate at the south of the Fenghuo Mountain region. The ERS-1 result in Figure 11a shows that the ground is stable and the displacement rate is mostly less than −5 mm/year. During 2004–2010, the InSAR results showed obvious ground deformation in the north. During 2015–2018, the surface deformation was more severe, and obvious deformations have occurred throughout the region. The largest was −20 mm/year during 2015–2018. Most of the QTR embankment showed a minor deformation rate.

#### 4.2.3. Tuotuohe

The average elevation of Tuotuohe region is about 4780 m. The ALT ranges from 1 to 4 m. Figure 12 shows the average deformation rate of Tuotuohe region during 1997–2018 from the ERS-1, ENVISAT and Sentinel-1A datasets. In the ERS-1 and ENVISAT deformation results (Figure 12a,b respectively), no obvious deformation area was found. From 2015 to 2018, serious deformation was found in this area, marked by red dotted ellipses. The largest deformation rate was over −20 mm/year per the Sentinel-1A results. Subsiding regions were found around the embankment of the QTR, which will be analyzed in the following section.

Figure 13 shows the time series displacement of the three selected points in this region. Because the number of the ERS was small, the timeseries displacements were analyzed for ENVISAT and sentinel-1A. The long-term subsidence was probably caused by melting of ground ice near the permafrost table [21]. The seasonal trend was remarkable, reflecting the effects of the thawing and freezing of the active layer. Points A, B and C exhibited the accumulative deformation less than 40 mm from 1997 to 2010. For the Sentinel-1A results, the time series displacement of the three points showed a similar seasonal trend, with the deformation rates of −8.5 mm/year, −20.1 mm/year and −11.9 mm/year, respectively. The accumulative displacement of Point B was 120 mm from 2015 to 2018. With time, deformations in parts of the Tuotuohe regions intensified. An increasing deformation trend was found in this region.

### 4.3. Deformation Analysis

#### 4.3.1. Deformation and Permafrost Thermal Regimes

In the last 2010s, the permafrost in QTP underwent serious degradation due to global warming. During the period from 1961 to 2007, the observed air temperatures over the QTP showed a rising trend, with a mean increasing rate of 0.037 °C/year [56]. Against the background of global warming, the air temperatures over the QTP continued to rise. The ground deformation was a manifestation of the degeneration of the permafrost. The mean annual ground temperature (MAGT) is often used for permafrost thermal regime mapping on a large scale. The MAGT is correlated with the elevation, local slope, soil properties, vegetation, location, and other factors [43]. Lu et al. [57] proposed a relationship model between MAGT and the elevation, latitude and slope aspects from 29 boreholes along the QTEC from Beiluhe to Fenghuo Mountain. The multi-correlation coefficient is significant with a value of 0.936. The study area in Lu et al. [57] is the same as our study site and the model is easy to application. So, the model is used to monitor the MAGT and evaluate the stability of the permafrost in our study site.

The modeled MAGT of the study site is shown in Figure 14. The modeled MAGTs were the lowest for the Fenghuo Mountain areas with the temperature of less than −2.0 °C and the highest for the river valley areas Tuotuohe with the temperature above 0 °C. For the Beiluhe basin areas, the relatively warm MAGTs ranged from −2.0 to 0 °C. The modeled MAGTs are consistent with the latest researches on MAGTs in QTP [58,59]. Comparing Figure 7 and Figure 14, we found that the subsiding regions are consistent with the ground with high MAGT value; the Tuotuohe and Beiluhe regions have experienced undergone serious ground deformation in recent years. High MAGTs would contribute to the acceleration of permafrost thawing and then increase the settlement of the ground.

#### 4.3.2. Deformation and Thaw Slumping

Thermokarst lakes have been developing along the QTEC as a result of increased human activity and ongoing climate warming [60]. The thermokarst lakes and thaw slumping have been observed more frequently in permafrost areas, such as the Beiluhe region and Fenghuo Mountain [61]. Thaw slumping has occurred near the embankments of QTR and QTH. In the regions with obvious ground deformation in our study area, some thaw slumps have been observed in the Tuotuohe region through the time series SAR amplitude maps.

Figure 15 shows the time-series amplitude maps of the Tuotuohe area from 2007 to 2018, the same area as that shown in Figure 12. At least three areas, marked as R1, R2 and R3, underwent thaw slumping throughout the whole observation period. By comparing Figure 12 and Figure 15, we found that from 2007 to 2018, the areas experiencing thaw slumping in the three regions have increased by 0.435, 0.679, 0.317 km^2^, respectively (Table 3). The distributions of thaw slumps areas are consistent with the ground deformation. The formation of thaw slumps may be initiated by several processes that expose ice-rich permafrost sloping terrain, which contributes to serious ground deformation [59]. The observed increase in areas of thermokarst lakes or thaw slumping regions indirectly validates our retrieved ground deformation result.

## 5. Discussion

We think that most of the embankments and foundations of the permafrost along QTEC are stable, but some sections are still experiencing obvious deformation. Based on the 20 years of InSAR observations, at least three regions have been identified as undergoing serious ground deformation, consistent with the previous studies in the QTP [30,31]. The ground deformation tends to expand. The embankments of QTR and QTH around Fenghuo Mountains should be reinforced as should points A, B and C near the Tuotuohe regions.

To evaluate the estimated results, the levelling measurement data should be collected. Because it is difficult for us to collect the levelling data in QTEC region, the estimated results could not be directly validated. However, several pieces of ground deformations evidences have been found in our field investigations that indirectly verifies the results. In the Beiluhe sections, visible fissures have been found in the QTR subgrade and alpine meadow region (Figure 10). We also compared our results with the previous studies in the QTP permafrost area (Table 1). In the Beiluhe area, several studies have been conducted on the deformation of permafrost using InSAR. Chen et al. [31] retrieved the ground deformation along the QTR in the Beiluhe area using C- and L-band small SAR interferometry. The estimated surface deformation rate along embankment ranges from −20 to +20 mm/year. Li et al., [32] monitored the surface deformation in the Beiluhe area using InSAR with ENVISAT images. The deformation velocity near the QTR embankment was larger than −10 mm/year. Similarly, our previous studies in the Beiluhe regions with TerraSAR-X ST mode images showed the similar deformation trends, with the deformation rate ranging from −20 to 0 mm/year [37]. Our retrieved ground deformation rate is consistent with those studies. The small differences between our findings and those reported by the previous studies are due to the following aspects: (1) different band SAR images and the InSAR processing method were used, which contributed to this difference, and (2) the observation periods were difference. Despite these case studies being conducted at different time periods, the gradual subsidence trends were all in the order of centimeters per year, similar to our reported subsidence trends. Most of the previous studies used the SAR images acquired before 2010. In this study, the latest ground deformation along QTEC were obtained.

There are three limitations in this study. Firstly, due to the complexity of the permafrost thawing and freezing process, monitoring the ground deformation using a physical equation was challenging. Linear [29,46], cubic polynomial [31], seasonal [33,36,49], and equation with climatic factors [35] and temperature [21,37,51] phase models have been used. These models have been applied successfully in some permafrost regions. Many other factors, such as vegetation coverage, soil moisture, snow cover, and solar radiation, should be considered in the future when monitoring the permafrost deformation.

Another limitation of the InSAR applications on permafrost regions is the temporal decorrelations [31,46]. The permafrost surface experiences dynamic environmental conditions and severe climate change from summer to winter season, which result in the dramatic temporal variations in the ground surface. Many studies used the SAR datasets acquired in the winter season [21,51] or use the L-band SAR images [62] to suppress the temporal decorrelations. Some methods and advanced methods have been proposed to solve this difficulty. Daout et al., 2017 used a PCA approach to help for the unwrapping in the decorrelated permafrost environment [39]. With the launches of satellites with long-wavelength SAR sensors such as ALOS-2, and the shortening of the satellite revisit cycle, and the development of advanced algorithm, InSAR technology (distributed scatterer interferometry, DSI) [63], the temporal decorrelation will be largely suppressed.

Last, comparing the estimated deformation rate and DEM error term, we found that they are the trade-offs for the ERS-1 images. We think at least two factors contribute to this. Firstly, a covariance exists between the temporal and perpendicular baseline, especially for ERS-1 data. The smaller the spatial perpendicular baseline, the higher the quality of the interferograms. The smaller the temporal baseline, the higher the quality of the interferograms. However, in the permafrost areas, the quality of the interferograms would be better between two images acquired in the same season with large temporal baseline and some interferograms with small temporal baseline are rejected due to serious temporal decorrelation. Secondly, in the permafrost region, the deformation may be correlated with the topography. Most of the subsiding areas are the plane regions (Beiluhe and Tuotuohe). In the mountainous areas, the deformation rate is small and stable. More SAR images with short revisit cycle are needed in the future research.

## 6. Conclusions

In this paper, we presented an application using the time-series InSAR technique with multisensory SAR datasets to monitor the permafrost ground deformation along the QTEC from 1997 to 2018. A deformation model combining a linear subsidence term and seasonal deformation term was adopted in the time-series InSAR method to exploit the permafrost ground deformation. Three deformation rate maps along a 100 km stretch of the QTEC were generated from 9 ERS-1, 39 ENVISAT, and 41 Sentinel-1A images. The three independent InSAR measurement results showed a consistent deformation trend and most of the ground surface along the QTEC was stable with the deformation rate ranging from −10 to 10 mm/year. The conclusions are summarized as follows:(1)Before the operation of the QTR, the QTEC from Wudaoliang to Tuotuohe was in stable with a deformation velocity of less than −5 mm/year from ERS-1 images. The embankment of the engineering structure was considered stable. The thawing and freezing of the active layer were the main deformation driving-forces. After the QTR started operation and the human activities increased, some sections of the QTEC were underwent obvious deformation, and the deformation has increased more recently.(2)From 2015 to 2018, obvious deformation was found in three areas: Beiluhe, southern of Fenghuo Mountain, and Tuotuohe, with the large deformation rates of over −20 mm/year. Real-time deformation monitoring must be conducted in these sections. The subsiding areas are consistent with the permafrost areas with large MAGTs.(3)This work demonstrated the potential of the time-series InSAR for the surveillance of the state of QTEC on a large scale. Interferometric decorrelation is still one of the problems for time-series InSAR monitoring of the ground deformation in permafrost region. With the proposed innovative methods and newly-launched SAR systems with shorter revisit cycles (Sentinel-1A/1B and TerraSAR-L), higher temporal sampling allows us to better characterize the ground deformation related to the process of permafrost thawing and freezing.

In future work we will focus on investigating the three-dimension deformation in permafrost regions using multiple satellites SAR images, and retrieving the geophysical parameters of permafrost such as the active layer thickness, on a larger scale.

## Figures and Tables

**Figure 1 sensors-19-05306-f001:**
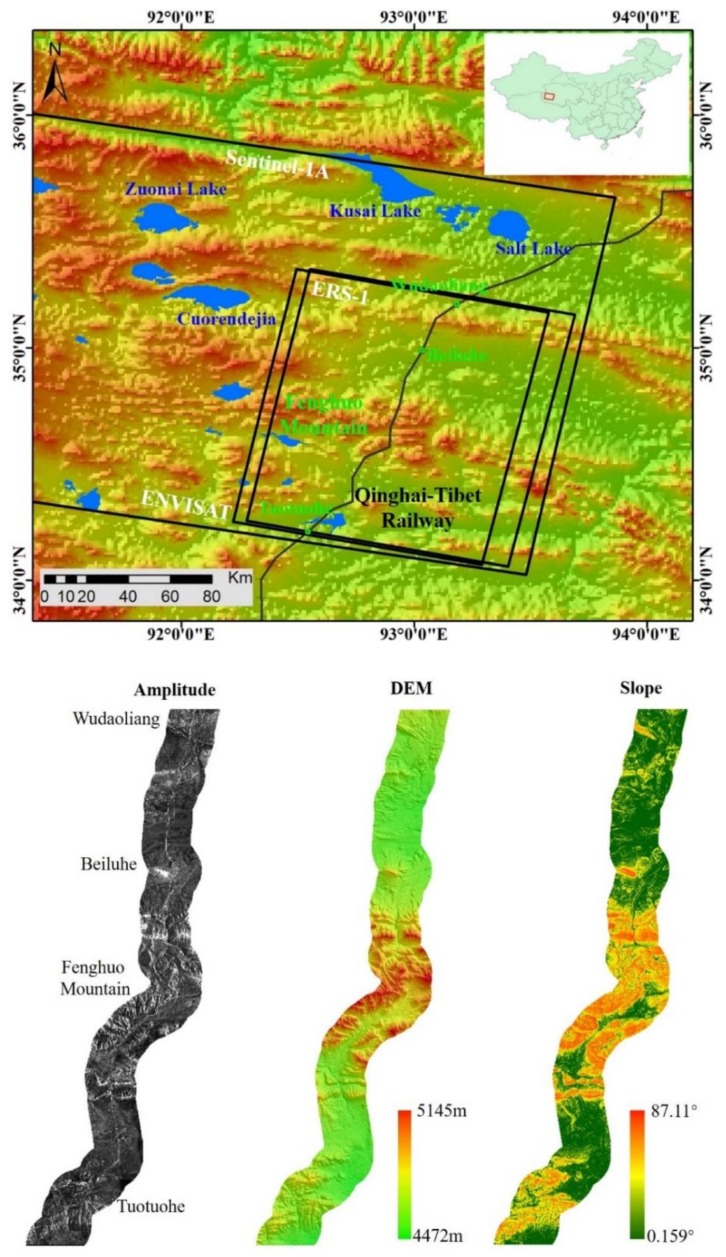
**Top**: Coverage of radar data stacks (black squares) on the shuttle radar topography mission (SRTM) digital elevation map (DEM) over the study area. The black lines show the Qinghai-Tibet Railway (QTR). The green points represent the railway station in the study area. The blue polygons are the large lakes. **Bottom**: the amplitude of Sentinel-1A, DEM, and slope of the selected Qinghai-Tibet Engineering corridor (QETC) section from Wudaoliang to Tuotuohe.

**Figure 2 sensors-19-05306-f002:**
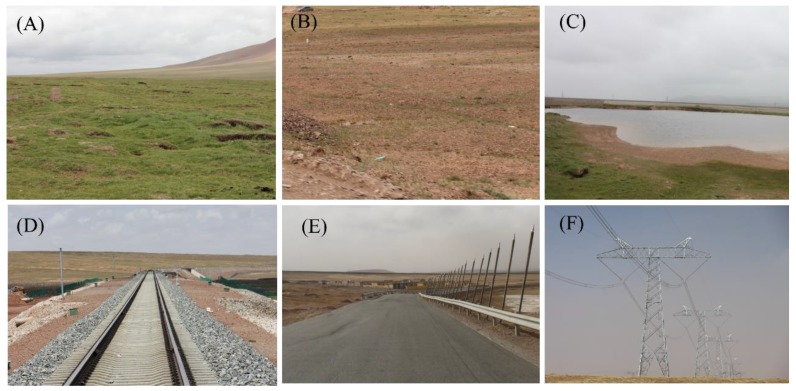
Field photos of the study area in August 2014. (**A**) alpine meadow, (**B**) alpine desert, (**C**) thermokarst lake, (**D**) QTR, (**E**) Qinghai-Tibet Highway (QTH), and (**F**) electric transmission power line.

**Figure 3 sensors-19-05306-f003:**
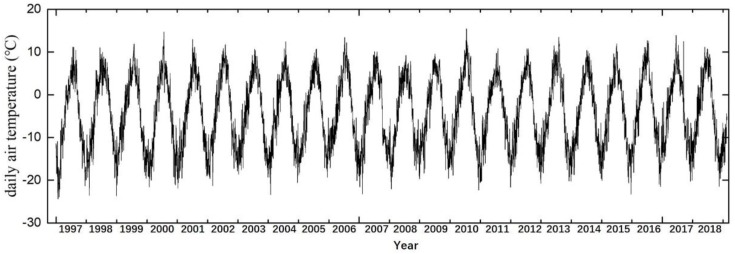
Daily air temperature in Wudaoliang from 1997 to 2018.

**Figure 4 sensors-19-05306-f004:**
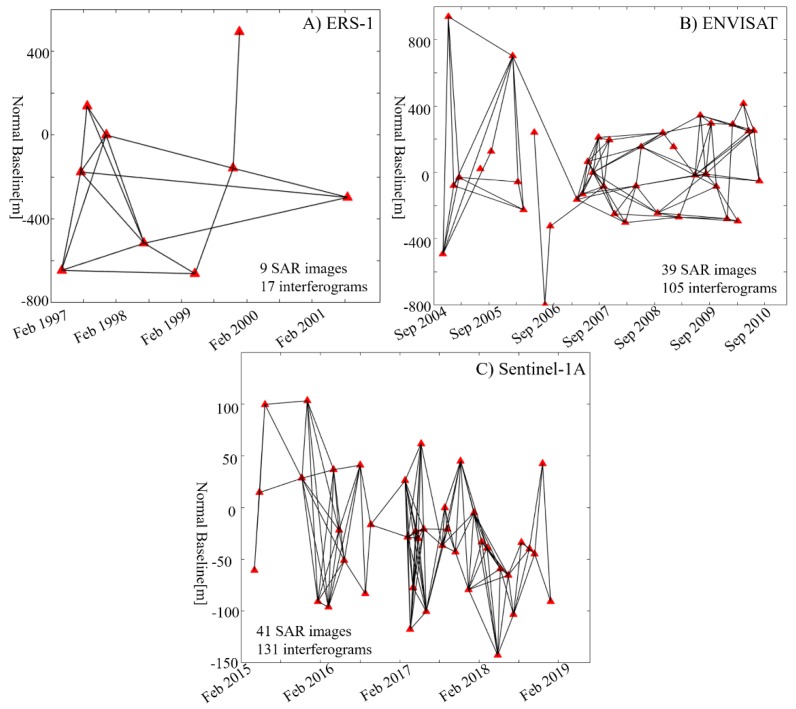
Generated interferometric pairs of (**A**) ERS-1, (**B**) ENVISAT, and (**C**) Sentinel-1A. All the lines represent the interferograms used to monitor the time-series ground deformation. All the points represent the SAR images.

**Figure 5 sensors-19-05306-f005:**
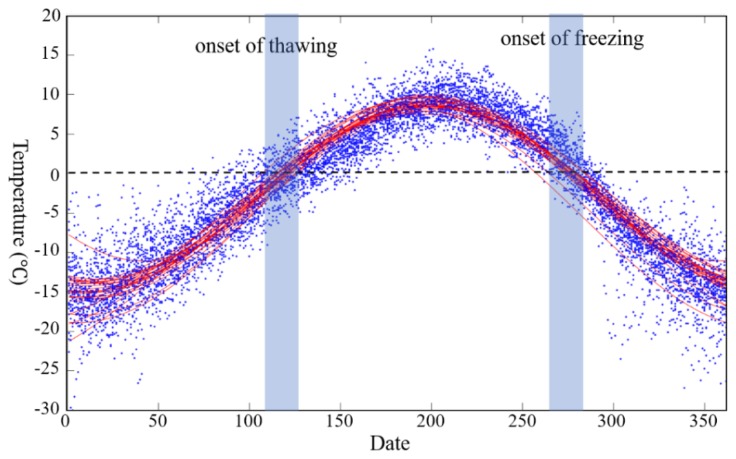
Seasonal pattern temperature of each year from 1997 to 2018 and the monitored models each year.

**Figure 6 sensors-19-05306-f006:**
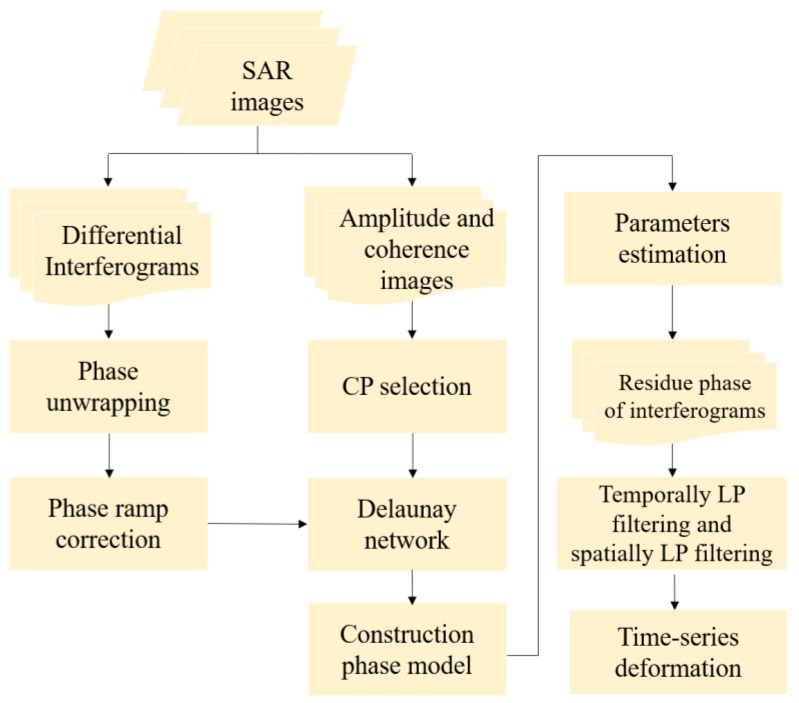
The flowchart of the time-series InSAR approach.

**Figure 7 sensors-19-05306-f007:**
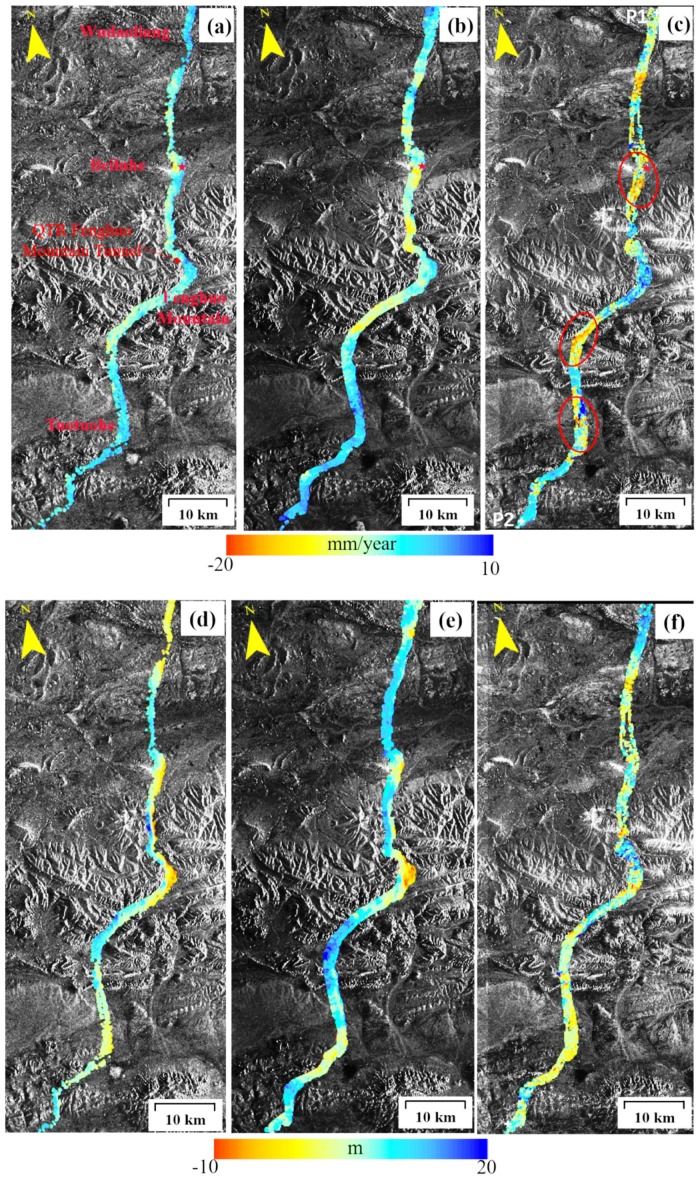
Estimated average ground deformation rate along the QTEC in (**a**) 1997–1999, (**b**) 2004–2010, and (**c**) 2015–2018 derived from the ERS-1, ENVISAT and Sentinel-1A data, respectively. The red star is the reference point. (**d**–**f**) are the corresponding estimated DEM error.

**Figure 8 sensors-19-05306-f008:**
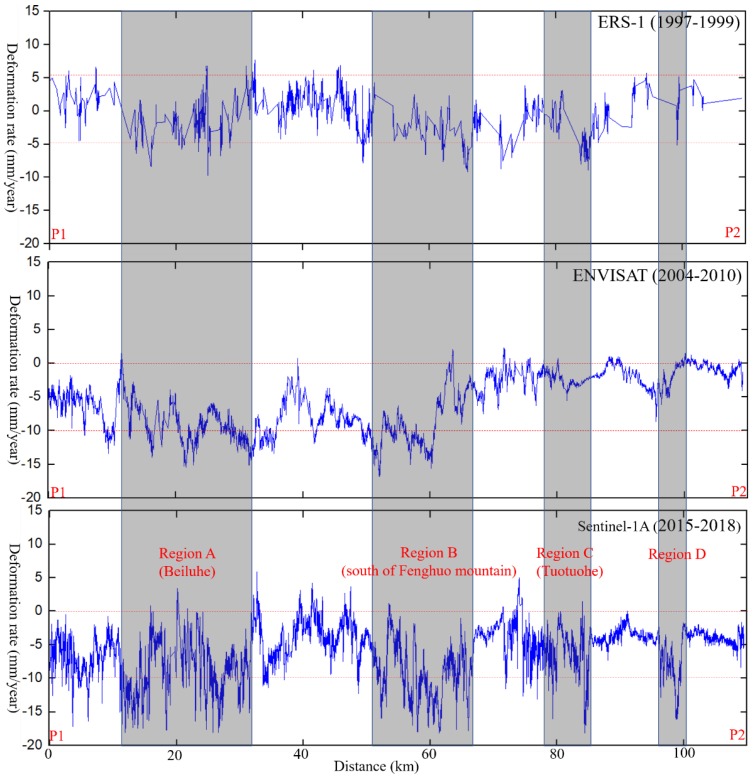
Deformation rate profile along the QTR, from point P1 to point P2 in Figure 7c.

**Figure 9 sensors-19-05306-f009:**
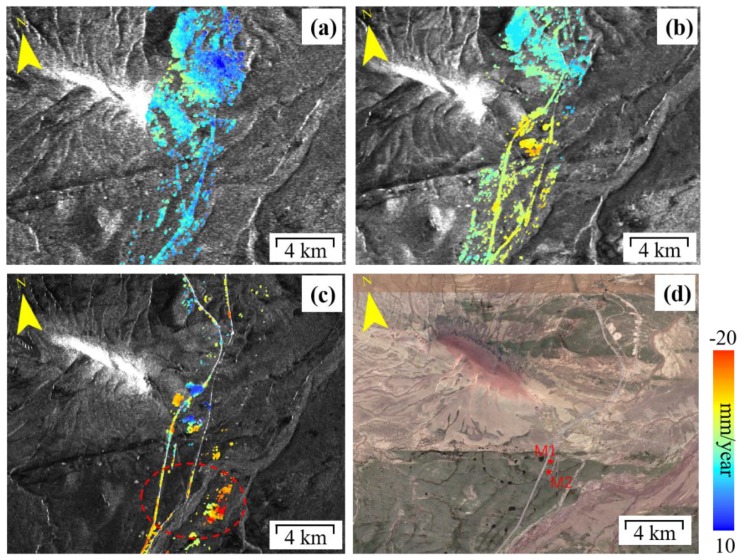
Permafrost ground deformation rate at the Beiluhe region from InSAR in 1997–2018. (**a**) ERS-1 1997–1999, (**b**) ENVISAT 2004–2010, (**c**) Sentinel-1A 2015–2018, and (**d**) the corresponding Google map.

**Figure 10 sensors-19-05306-f010:**
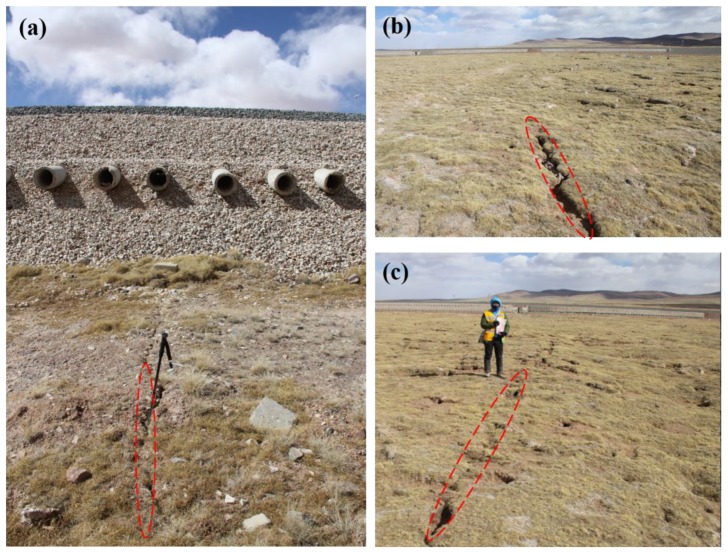
Field photos with surface cracks and fissures in the Beiluhe region (March 2015). (**a**) Photo taken at the point M1 in Figure 9d. (**b**) and (**c**) Photos taken at the point M2 in Figure 9d.

**Figure 11 sensors-19-05306-f011:**
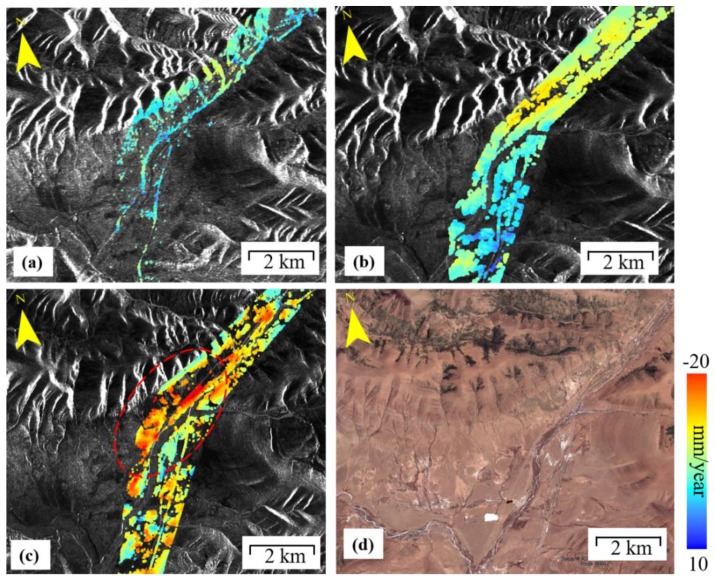
Permafrost ground deformation rate in the south of Fenghuo Mountain from InSAR during 1997–2018. (**a**) ERS-1 1997–1999, (**b**) ENVISAT 2004–2010, (**c**) Sentinel-1A 2015–2018, and (**d**) the corresponding Google map.

**Figure 12 sensors-19-05306-f012:**
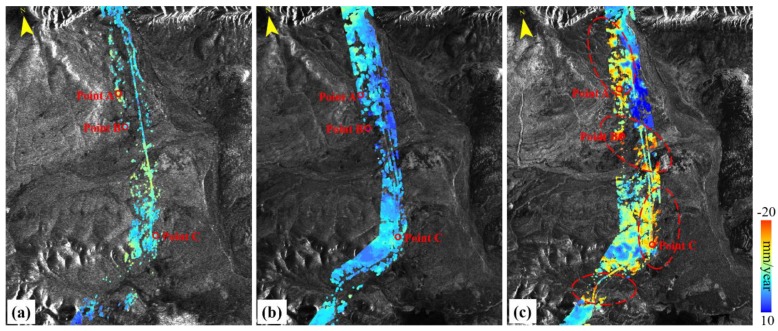
Permafrost ground deformation rate in the Tuotuohe region from InSAR during 1997–2018. (**a**) ERS-1 1997–1999, (**b**) ENVISAT 2004–2010, (**c**) Sentinel-1A 2015–2018.

**Figure 13 sensors-19-05306-f013:**
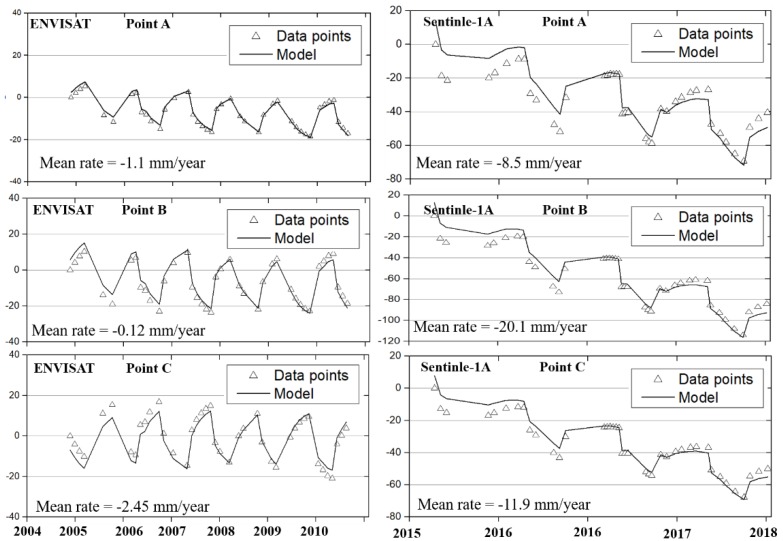
Time-series deformation of Point A, B, and C from 2004–2010 (ENVISAT) and 2015–2018 (Sentinel-1A). The hollow triangle indicates the time-series displacement using the InSAR method, and the black polylines denote the modeled deformation.

**Figure 14 sensors-19-05306-f014:**
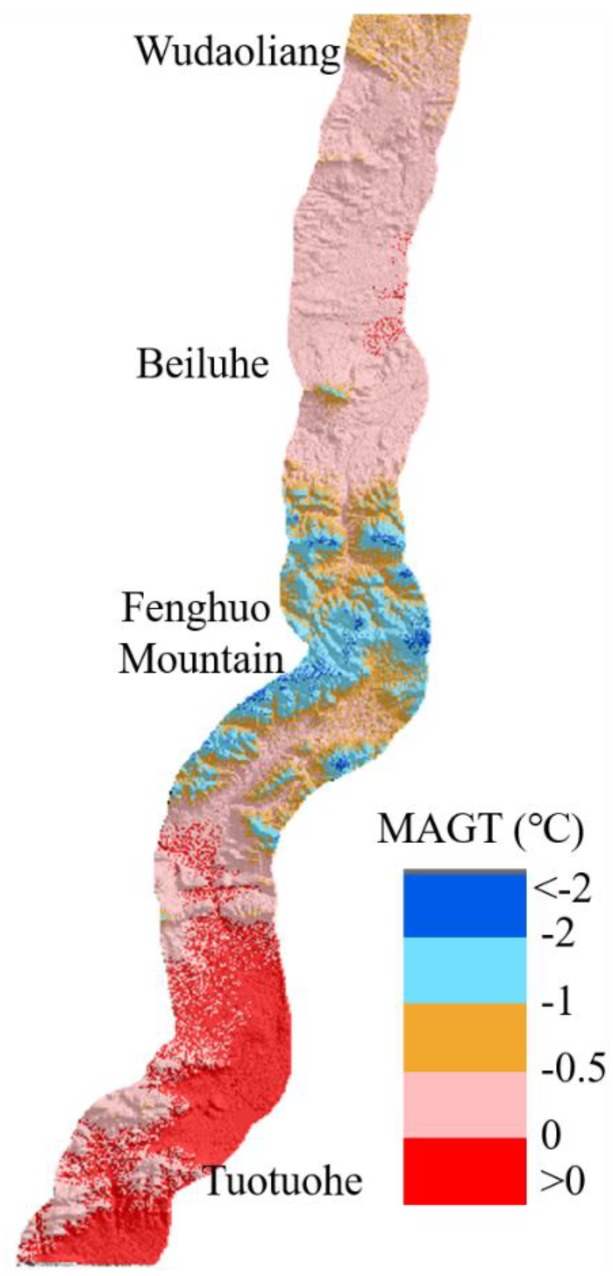
Map of the modeled mean annual ground temperature (MAGT) along the QTEC from Wudaoliang to Tuotuohe.

**Figure 15 sensors-19-05306-f015:**
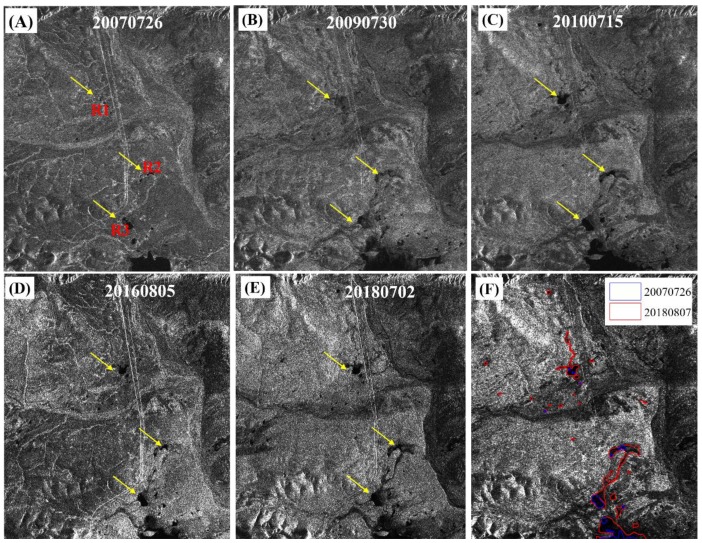
The time-series amplitude maps of Tuotuohe area, the same location as that shown in Figure 9: (**a**) 26 July 2007, (**b**) 30 July 2009, (**c**) 15 July 2010, (**d**) 5 August 2016, (**e**) 2 July 2018, and (**f**) water regions between 26 July 2007 and 7 August 2018. The blue polygon indicates 26 July 2007 and the red polygon indicates 7 August 2018. The yellow arrows indicate the water regions: R1, R2, and R3.

**Table 1 sensors-19-05306-t001:** Permafrost deformation studies in the Qinghai-Tibet Plateau (QTP) using synthetic aperture radar interferometry (InSAR) technologies.

Study Areas	InSAR Method	SAR Dataset	Observation Period	Deformation Rate (mm/year)	References
Beiluhe	PSI	ENVISAT	August 2003–May 2007	−20 to 3	[29]
Beiluhe	IPTA and SBAS	ALOS-1 and ENVISAT	November 2004–December 2010	−20 to 20	[30]
Beiluhe	SBAS	ALOS-1	June 2007–December 2010	−20 to 20	[31]
Beiluhe	SBAS	ENVISAT	April 2003–July 2010	−16 to 2	[32]
Tanggula	PSI	ENVISAT	February 2007–September 2009	−10 to 0	[33]
Yangbajing	MTInSAR	TerraSAR-X	December 2011–November 2012	−30 to 10	[34]
Yangbajing	SBAS	ENVISAT	May 2007–September 2010	−50 to 10	[35]
Wudaoliang	SBAS	ALOS-1	May 2007–March 2009	−2 to 0	[36]
Beiluhe	MTInSAR	TerraSAR-X	July 2014–March 2017	−20 to 0	[37]
Wudaoliang-Fenghuo Mountain	MTInSAR	Sentinle-1A	November 2017–December 2018	—	[38]
Northwestern Tibet	NSBAS (new small baseline subset)	ENVISAT	2003–2011	−4 to 4	[39]

**Table 2 sensors-19-05306-t002:** SAR image numbers and parameters used in this study.

Sensors	Start and End Date	Acquisitions (n)	Incidence Angle (°)	Polarization	Pixel Spacing/Range (m)	Pixel Spacing/Azimuth (m)
ERS-1	1997-04-24 to 1999-12-30	9	19.3~26.5	VV	7.9	3.9
ENVISAT	2004-11-18 to 2010-07-15	39	18.6~26.2	VV	7.8	4
Sentinel-1A	2015-04-13 to 2018-12-17	40	30.7~37.6	VV	5	20

**Table 3 sensors-19-05306-t003:** Areas of regions R1, R2 and R3 between 2007-07-26 and 2018-08-07.

Region	Areas (km^2^)		
	26 July 2007	7 August 2018	Change
R1	0.023	0.458	0.435
R2	0.068	0.747	0.679
R3	0.244	0.561	0.317

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
