# Peer review of "Permafrost Deformation Monitoring Along the Qinghai-Tibet Plateau Engineering Corridor Using InSAR Observations with Multi-Sensor SAR Datasets from 1997–2018"

_sensors, 2019, doi:10.3390/s19235306_

Round 1
Reviewer 1 Report
I still have concerns regarding the decomposition of the data from equation 4, as the inverse problem is underdetermined due to the complexity of the seasonal degree-day model and the trade-offs with the DEM errors. Thanks to the modifications from the first version of the manuscript, we can now see that the DEM error terms are not only very high for ERS data, but has also strong impact in the Sentinel time series model (e.g in Fig. 13), while Sentinel data have very small geometrical baselines and are therefore supposed to have low DEM errors. I have also noticed that the authors have changed the decomposition approach for the ERS data set (with the poorest data sampling) now including a sinusoidal function instead of the complex seasonal degree-day function and have removed most of the time series to only show few in the Fig.13 at a single place (Tuotuohe) since I asked in my previous review to also show the modeled time series. My belief is that 1) a sinusoidal function should be applied to all data sets in order to not bias the estimation of the long-term trend, 2) the proposed fit from eq. 4 is wrong at several areas as there is a spatial variability of the thawing and freezing onset due to various factors and as there are other factors that control the FT cycles state of the active layer than the temperature. Again, see Daout et al., 2017,” Large scale InSAR monitoring of permafrost freeze-thaw cycles on the Tibetan Plateau” or Rouyet et al., 2019, “Seasonal dynamics of a permafrost landscape, Adventdalen, Svalbard, investigated by InSAR” or “Dini et al., Classification of slope processes based on multitemporal DInSAR analyses in T the Himalaya of NW Bhutan, RSE, 2019” who demonstrate with an InSAR time series approach that the amplitude but also the timing of the FT deformation depends on the shadowing, the sediment type or the local topography slopes. In my opinion, it is, therefore, wrong to force the SBAS model inversion from Eq 4 with such parametric form depending on one a single temperature time series at a single place with a fixed freezing and thawing onsets. However, as the authors now show the DEM errors in map view for the 3 datasets and as the time series models are available in Fig. 13 (but not anymore for ERS), the reader has all elements to judge by themselves and I am not expecting the authors to do any changes.
I am also particularly disturbed by the wording, found in more than one place in the paper (e.g. in the abstract, l.65 “This research indicates that InSAR technique could play a significant role in monitoring the ground deformation along QTEC. “ or in the conclusion line 473 “This work demonstrates the promise of the time-series InSAR for surveillance the state of QTEC in large scale “, check where else). They claim that their paper brings about novel findings such as the ability to show trends and monitor permafrost-related instabilities. They also claim that l. 447-448. “Many researches only use the SAR datasets acquired only in winter season to suppress the temporal decorrelations”, or that “l. 430 “Due to the complexity of the permafrost thawing and freezing process, it is difficult to monitor the ground motion using physical equation” or that “The literatures have only focused on deformation monitoring in a short period of time”. This is simply not true and it would be good and intellectually honest to recognise the efforts of other authors as well. In recent years, Rouyet et al (2019), Dini et al., (2019) show how their processing and data revealed similar processes using Terrasar-X, Sentinel or ALOS data. Daout et al., 2017, analyzed the temporal and spatial variations of the permafrost ALT on the Tibetan Plateau over a 60,000 km2 region and introduced a method, based on a PCA approach, to help for the unwrapping in such decorrelated environment. In my view, it is not licit to claim novel findings without acknowledging the work of other authors. The authors should also be careful when they say l. 451 “With the launches of satellite with long-wavelength SAR sensor such as ALOS-2 and shortening of the satellite revisit cycle, the temporal decorrelation will be largely suppressed. ”, or l. 476 “With the proposed innovative methods and newly-launched SAR systems with shorter revisit cycle (Sentinel- 1A/1B and TerraSAR-X-TanDEM-X), this limitation will be overcome to some extent”, as they also use data with very good temporal sampling in their study (eg. Sentinel-1 data) and as other studies, like in Daout et al., 2017, have proposed method to face the unwrapping difficulties in permafrost environment with data with lower temporal sampling.
Author Response
I still have concerns regarding the decomposition of the data from equation 4, as the inverse problem is underdetermined due to the complexity of the seasonal degree-day model and the trade-offs with the DEM errors. Thanks to the modifications from the first version of the manuscript, we can now see that the DEM error terms are not only very high for ERS data, but has also strong impact in the Sentinel time series model (e.g in Fig. 13), while Sentinel data have very small geometrical baselines and are therefore supposed to have low DEM errors.
I have also noticed that the authors have changed the decomposition approach for the ERS data set (with the poorest data sampling) now including a sinusoidal function instead of the complex seasonal degree-day function and have removed most of the time series to only show few in the Fig.13 at a single place (Tuotuohe) since I asked in my previous review to also show the modeled time series.
Response: Thanks for your comments. I am sorry for misleading you.
In this paper, we don’t use a fixed freezing and thawing onsets. In the section 3.1, the calculation of the ADDT and ADDF is given. A sinusoidal function is used to identify the onsets of each year. Through our result, we found that the onsets vary from each year (Table 1).
Table 1 Onsets of thawing and freezing season from 1997-2018
Year |
Onset of thawing season (month-day) |
Onset of freezing season (month-day) |
R2 |
1997 |
5-10 |
9-26 |
0.91 |
1998 |
5-3 |
10-9 |
0.91 |
1999 |
4-21 |
9-19 |
0.82 |
2000 |
5-8 |
10-7 |
0.94 |
2001 |
5-10 |
10-11 |
0.91 |
2002 |
5-3 |
10-7 |
0.91 |
2003 |
5-5 |
10-11 |
0.9 |
2004 |
5-1 |
10-4 |
0.89 |
2005 |
5-5 |
10-8 |
0.91 |
2006 |
5-7 |
10-7 |
0.88 |
2007 |
5-4 |
10-10 |
0.87 |
2008 |
5-8 |
10-5 |
0.91 |
2009 |
4-30 |
10-11 |
0.89 |
2010 |
4-27 |
10-10 |
0.9 |
2011 |
5-6 |
10-11 |
0.91 |
2012 |
5-4 |
10-7 |
0.92 |
2013 |
4-29 |
10-4 |
0.89 |
2014 |
5-4 |
10-8 |
0.91 |
2015 |
4-29 |
10-12 |
0.9 |
2016 |
5-1 |
10-15 |
0.92 |
2017 |
5-4 |
10-15 |
0.87 |
2018 |
4-30 |
10-8 |
0.9 |
As your suggestion in the previous review, a deformation model without a seasonal term was used for ERS-1 datasets. The sinusoidal function deformation phase model is not used for the three SAR images.
My belief is that
1) a sinusoidal function should be applied to all data sets in order to not bias the estimation of the long-term trend,
2) the proposed fit from eq. 4 is wrong at several areas as there is a spatial variability of the thawing and freezing onset due to various factors and as there are other factors that control the FT cycles state of the active layer than the temperature.
Response:
As you said, the temporal nature of permafrost thawing and freezing is complex and is related with several factors, such as temperature, soil moisture, soil property, etc. Moreover, there are intermediate cases with more complex temporal variability. For instance, a transient layer between the active layer and the ice-rich permafrost would serve as a buffer zone to delay thawing in the ice-rich permafrost (Liu et al., 2015). Temperature is one of the most important factors. So, many studies use the temperature to monitor the permafrost deformation. Many studies used the sinusoidal function phase model to retrieve the permafrost ground deformation (Daout et al., 2017; Dini et al., 2019). The time delay between the temperature maxima and deformation maxima has been found. The thawing and freezing process of permafrost is not a sinusoidal function. The freezing season is longer that that of thawing season. At the end of freezing season, the ground is relative stable. Figure 1 shows a relatively simple scenario (Liu et al., 2015; Chen et al., 2018). Some studies use the Stefan model to describe the ground deformation based on the measurement temperature (Liu et al., 2012). It is believed that this model is relative reasonable because the deformation over permafrost is also accumulated little by little. We also have used this model to retrieve permafrost deformation and active layer thickness in QTP Beiluhe permafrost region (Wang et al., 2017; 2018). Both two models have their advantages. The sinusoidal function model is good for simulating seasonality. The phase model based on Stefan have considered the accumulate climatic factor.
Figure 1. schematic diagrams of the surface elevation change on permafrost region.
Reference:
Chen, J., Liu, L., Zhang, T., Cao, B., & Lin, H. (2018). Using Persistent Scatterer Interferometry to Map and Quantify Permafrost Thaw Subsidence: A Case Study of Eboling Mountain on the Qinghai‐Tibet Plateau. Journal of Geophysical Research: Earth Surface, 123(10), 2663-2676.
Daout, S.; Doin, M. P.; Peltzer, G.; Socquet, A.; Lasserre, C.. Large‐scale InSAR monitoring of permafrost freeze‐thaw cycles on the Tibetan Plateau. Geo. Rese. Lett. 2017, 44(2), 901-909.
Dini, B., Daout, S., Manconi, A., & Loew, S. (2019). Classification of slope processes based on multitemporal DInSAR analyses in the Himalaya of NW Bhutan. Remote Sensing of Environment, 233, 111408.
Liu, L.; Schaefer, K.; Zhang, T.; Wahr, J.. Estimating 1992–2000 average active layer thickness on the Alaskan North Slope from remotely sensed surface subsidence. J. Geophys. Res-Earth 2012, 117(F1).
Liu, L., Schaefer, K. M., Chen, A. C., Gusmeroli, A., Zebker, H. A., & Zhang, T. (2015). Remote sensing measurements of thermokarst subsidence using InSAR. Journal of Geophysical Research: Earth Surface, 120(9), 1935-1948.
Wang, C.; Zhang, Z.; Zhang, H.; Wu, Q.; Zhang, B.; Tang, Y.. Seasonal deformation features on Qinghai-Tibet railway observed using time-series InSAR technique with high-resolution TerraSAR-X images. Remote Sens. Lett. 2017, 8(1), 1-10.
Wang, C.; Zhang, Z.; Zhang, H.; Zhang, B.; Tang, Y.; Wu, Q.. Active Layer Thickness Retrieval of Qinghai–Tibet Permafrost Using the TerraSAR-X InSAR Technique. IEEE J-STARS 2018, 11(11), 4403-4413.
Zhao, R.; Li, Z. W.; Feng, G. C.; Wang, Q. J.; Hu, J.. Monitoring surface deformation over permafrost with an improved SBAS-InSAR algorithm: With emphasis on climatic factors modeling. Remote Sens. Environ. 2016, 184, 276-287.
Again, see Daout et al., 2017,” Large scale InSAR monitoring of permafrost freeze-thaw cycles on the Tibetan Plateau” or Rouyet et al., 2019, “Seasonal dynamics of a permafrost landscape, Adventdalen, Svalbard, investigated by InSAR” or “Dini et al., Classification of slope processes based on multitemporal DInSAR analyses in T the Himalaya of NW Bhutan, RSE, 2019” who demonstrate with an InSAR time series approach that the amplitude but also the timing of the FT deformation depends on the shadowing, the sediment type or the local topography slopes. In my opinion, it is, therefore, wrong to force the SBAS model inversion from Eq 4 with such parametric form depending on one a single temperature time series at a single place with a fixed freezing and thawing onsets.
However, as the authors now show the DEM errors in map view for the 3 datasets and as the time series models are available in Fig. 13 (but not anymore for ERS), the reader has all elements to judge by themselves and I am not expecting the authors to do any changes.
I am also particularly disturbed by the wording, found in more than one place in the paper (e.g. in the abstract, l.65 “This research indicates that InSAR technique could play a significant role in monitoring the ground deformation along QTEC. “ or in the conclusion line 473 “This work demonstrates the promise of the time-series InSAR for surveillance the state of QTEC in large scale “, check where else). They claim that their paper brings about novel findings such as the ability to show trends and monitor permafrost-related instabilities. They also claim that l. 447-448. “Many researches only use the SAR datasets acquired only in winter season to suppress the temporal decorrelations”, or that “l. 430 “Due to the complexity of the permafrost thawing and freezing process, it is difficult to monitor the ground motion using physical equation” or that “The literatures have only focused on deformation monitoring in a short period of time”. This is simply not true and it would be good and intellectually honest to recognise the efforts of other authors as well. In recent years, Rouyet et al (2019), Dini et al., (2019) show how their processing and data revealed similar processes using Terrasar-X, Sentinel or ALOS data. Daout et al., 2017, analyzed the temporal and spatial variations of the permafrost ALT on the Tibetan Plateau over a 60,000 km2 region and introduced a method, based on a PCA approach, to help for the unwrapping in such decorrelated environment.
In my view, it is not licit to claim novel findings without acknowledging the work of other authors. The authors should also be careful when they say l. 451 “With the launches of satellite with long-wavelength SAR sensor such as ALOS-2 and shortening of the satellite revisit cycle, the temporal decorrelation will be largely suppressed. ”, or l. 476 “With the proposed innovative methods and newly-launched SAR systems with shorter revisit cycle (Sentinel- 1A/1B and TerraSAR-X-TanDEM-X), this limitation will be overcome to some extent”, as they also use data with very good temporal sampling in their study (eg. Sentinel-1 data) and as other studies, like in Daout et al., 2017, have proposed method to face the unwrapping difficulties in permafrost environment with data with lower temporal sampling.
Response: Thanks for your comments. in the new version, we have changed the inappropriate sentences.

Reviewer 2 Report
A thorough survey and continuous monitoring of the ground deformation for QTEC and larger QTP is of great importance for permafrost study as well as local infrastructure. The study tried to complete this task by mapping deformation time series using a number of InSAR observations, but a few key elements in methods and validations are missing:
Table 2: I wonder why VH polarization was selected for Sentinel SAR? Why just one incidence angle was listed, which is supposed to vary with pixels? How did you determine the baselines? Which software package was used for the study? Section 3.3: why not use global freeze thaw datasets for calculating ADDT or ADDF? Section 3.4.2: Do you have an independent evaluation of the approach to remove atmospheric phase tested under variant atmosphere conditions (e.g. high and low water vapor levels)? Section 4: it is unclear if the deformation data are consistent among different sensor records; and if sensor-related biases exist. Quantified evaluations are strongly recommended; Line 436-446: the logic flow here is confusing; There are numerous typos in the paper. Just name a few: line 44 “team”; Line 71 “can been”; Line 90 “thermalkarst”; Line 152 “of greater”; line 160 “combing”; line 220 “optically estimated”.
Author Response
A thorough survey and continuous monitoring of the ground deformation for QTEC and larger QTP is of great importance for permafrost study as well as local infrastructure. The study tried to complete this task by mapping deformation time series using a number of InSAR observations, but a few key elements in methods and validations are missing:
Table 2: I wonder why VH polarization was selected for Sentinel SAR? Why just one incidence angle was listed, which is supposed to vary with pixels? How did you determine the baselines? Which software package was used for the study?
Response: Thanks for your careful review. It is a mistake. We used the VV polarization Sentinel-1A images in this paper. We have changed it in the new version. (please see line 161)
You are right. The incidence angle varies with pixels. In the list, we use the central incidence angle. In the new version, the incidence angle ranges of the three datasets are given. (please see line 161)
In this study, the interferograms and the deformation time-series were generated using our own software.
Section 3.3: why not use global freeze thaw datasets for calculating ADDT or ADDF?
Response: Thanks for your comments.
We have downloaded the Global Annual Freezing and Thawing Indices from the website [1]. This dataset contains the total annual freezing and thawing indices with a spatial resolution of 0.5 degrees latitude by 0.5 degrees longitude. However, we want to identify the daily ADDT and ADDF based on the acquisition of the SAR images. So, we didn’t use the global annual freezing and thawing indices datasets and used the daily air temperature in Wudaoliang weather station from 1997 to 2018.
Reference
[1]https://catalog.data.gov/dataset/global-annual-freezing-and-thawing-indices-version-1
Section 3.4.2: Do you have an independent evaluation of the approach to remove atmospheric phase tested under variant atmosphere conditions (e.g. high and low water vapor levels)?
Response: Thanks for your comments.
Two typical interferograms with obvious orbit errors and topographic related phase errors are shown in Figure 1. It would produce significant artifacts in the final displacement result if those phase ramps are not removed. Figure 1 (b) and (f) indicate the corrected interferograms using the phase ramps correction model (2). Most of the phase ramps (orbit error, topographic related atmospheric error) have been removed and the deformation signal are retained and observed in the corrected interferograms. The phase ramps correction model has been used in many studies (Sun et al., 2015; Zhang et al., 2014; Zhang et al., 2018).
After that, all the identified CPs are connected based on the Delaunay triangulation network. Then, the differential phase of all the edges in the triangulation network are calculated, which is beneficial to further remove atmospheric delay (Ferretti et al., 2002).
It is assuming that the atmospheric phase can be mostly removed based on the above steps. In this paper, independent evaluation of the approach to remove atmospheric phase tested under variant atmosphere conditions (e.g. high and low water vapor levels) have not been provide.
Figure 1. Examples of the interferograms before and after correction. (a): interferogram 20170225-20170414 before correction; (b): interferogram 20170225-20170414 after correction; (c): interferogram 20150413-20150507 before correction; (d): interferogram 20150413-20150507 after correction; (please see pdf file)
Reference:
Ferretti, A.; Prati, C.; Rocca, F.. Nonlinear subsidence rate estimation using permanent scatterers in differential sar interferometry. IEEE Trans. Geosci. Remote Sens. 2002, 38(5), 2202-2212.
Sun, Q.; Zhang, L.; Ding, X.L.; Hu, J.; Li, Z.W.; Zhu, J.J. Slope deformation prior to Zhouqu, China landslide from InSAR time series analysis. Remote Sens. Environ. 2015, 156, 45–57.
Zhang, L.; Ding, X.; Lu, Z.; Jung, H.S.; Hu, J.; Feng, G. A novel multitemporal InSAR model for joint estimation of deformation rates and orbital errors. IEEE Trans. Geosci. Remote Sens. 2014, 52, 3529–3540.
Zhang, Z. , Wang, C. , Wang, M. , Wang, Z. , & Zhang, H. . (2018). Surface deformation monitoring in zhengzhou city from 2014 to 2016 using time-series insar. Remote Sensing, 10(11).
Section 4: it is unclear if the deformation data are consistent among different sensor records; and if sensor-related biases exist. Quantified evaluations are strongly recommended;
Response: Thanks for your comment.
Firstly, through several field investigations many subsidence phenomena were found in the study areas (pictures of Beiluhe in Figure 10), which could verify our monitoring subsidence results indirectly. On the other hand, in the Tuotuohe region the water region change has been obtained and analyzed, which is related to the thawing of the permafrost.
In our results, the deformation rate is -20 to +10 mm/year. Chen et al. (2013) retrieved the ground deformation along QTR in Beiluhe area using C- and L-band small SAR interferometry. The estimated surface motion rate along embankment ranges from -20 to +20 mm/year. Li et al., (2015) monitored the surface deformation in Beiluhe area using InSAR with ENVISAT images. The deformation velocity near the QTR embankment is larger than -10 mm/year. Similar, our previous studies in Beiluhe regions with TerraSAR-X ST mode images show the similar deformation trends, with the motion rate ranges from -20 to 0 mm/year (Wang et al., 2018). The small differences between this paper and the previous studies due to the following aspects: 1) different band SAR images and InSAR processing method are used, which contributed to this difference; 2) the observation periods are difference, which would be another factor. We note that despite that these case studies are conducted at different time periods, the gradual subsidence trends are all on the order of centimeters per year, similar with our reported subsidence trends. It should be noticed that most of the previous studies used the SAR images acquired before 2010. In this paper, the latest ground deformation along QTEC have been obtained.
The leveling measurement is the most effective means of verification. In our further work, we will collect the ground measurement data to validate our result directly.
Reference:
Chen, F.; Lin, H.; Zhou, W.; Hong, T.; Wang, G.. Surface deformation detected by ALOS PALSAR small baseline SAR interferometry over permafrost environment of Beiluhe section, Tibet Plateau, China. Remote Sens. Environ. 2013, 138, 10-18.
Li, Z.; Tang, P.; Zhou, J.; Tian, B.; Chen, Q.; Fu, S. Permafrost environment monitoring on the Qinghai-Tibet Plateau using time series ASAR images. Int. J. Digit. Earth 2015, 8(10), 840-860.
Wang, C.; Zhang, Z.; Zhang, H.; Zhang, B.; Tang, Y.; Wu, Q.. Active Layer Thickness Retrieval of Qinghai–Tibet Permafrost Using the TerraSAR-X InSAR Technique. IEEE J-STARS 2018, 11(11), 4403-4413.
Line 436-446: the logic flow here is confusing;
Response: Thanks for your comments. This paragraph mainly discusses the trade-offs between deformation and DEM error term. We have changes location of this paragraph. (please see line 558)
There are numerous typos in the paper.
Just name a few:
line 44 “team”;
Line 71 “can been”;
Line 90 “thermalkarst”;
Line 152 “of greater”;
line 160 “combing”;
line 220 “optically estimated”.
Response: thanks for your careful review. The above and other typos have been corrected in the new version.

Reviewer 3 Report
The manuscript in its current state is not acceptable for publication at Sensors. Following are the main issues that would need to be addressed for the study to be considered for publication:
Extensive revisions of the English expression is required – the text is barely understandable in several places. Examples are provided for the abstract in the minor comments section below to show the extent of the revision required.
The title is too long, too technical.
The text needs to be re-organized in several places as methodological details are now in the result and even the discussion section and discussion elements are in the result section (see comments below). The discussion is more or less an extension of the result section.
The analysis could have been deepened much further without too much efforts: a better analysis of the spatio-temporal patterns of permafrost deformation would have been very valuable. For instance, a more straight forward assessment of the impact of the QTR construction on permafrost deformation could have been done by comparing in a more quantitative way deformation rates over time. Furthermore, similarly to the averaged deformation rates (over time) presented in figure 8, time series of averaged deformation rates (across space) would have been very nice to see, in relation to ground or air temperature time series.
The terminology needs to be homogenized (e.g. permafrost motion / deformation rate).
Some methodological details are missing (e.g. the reasoning behind the definitions of various regions for the analysis).
Too many acronyms are used, making the reading difficult.
Labeling of locations in the map figures need to be bigger.
MINOR COMMENTS
Abstract:
L15: what do you mean by “the embankment of the permafrost engineering”.
L16: “the freezing and thawing effects of active layer” à “the freezing and thawing seasonal cycles”
L17: “reveal” à assess
L18: “motions” à deformation
L18: “from Wudaoliang to Tuotuohe section” à useless detail.
L19: are these “linear subsidence trends” based on interannual variations? If so, then change it to “interannual linear subsidence trends”.
L20: “introduced” à presented
L20/21: “Experimental results show that the ground deformation rate along QTEC ranges from -5 to +5 mm/year during 1997-1999 observation period derived from ERS-1 data.” à Analysis of the ERS-1 data show ground deformation along QTEC ranging from -5 to +5 mm/year during the 1997-1999 observation period.
L22: “velocity” à rate. Be consistent in the terms you use.
L22: “primarily” à no needed
L24: reformulation needed, e.g. “Significant ground deformation was detected in three sections of the corridor using Sentinel-1A and InSAR imagery.”
L26/27: reformulation needed, e.g. “Ground deformation was associated with the development of thaw slumps in one of the three sections.”
Introduction:
L33: “is one of the most significant factors affecting” à “has the potential to affect”
L64: If it is already known that InSAR is a promising technique, it doesn’t need to be the conclusion of your abstract.
Table 1 is not useful at this point. Furthermore all the acronyms for the InSAR methods need to be explained in the caption.
L68: what does this expression “the relationship between permafrost and Qinghai-Tibet Plateau Engineering” mean? Do you mean the consequences of permafrost deformation on the Qinghai-Tibet Plateau Engineering development?
L77: PS and SBAS: explain these acronyms.
Study area:
L90: “thermalkarst” à thermokarst
L90 and L95: references needed.
Figure 1: explain SRTM and include a reference for the DEM map. The Amplitude is from which product?
L93: mention the panels in Figure 2 associated with each land cover.
L104: what does this means??? “Due to the constructions of those engineering, the statements of the permafrost have been broken.”
L105: replace motion with deformation please.
Figure 3 no needed.
L132: “all the SAR images are co-registrated” à what does this mean?
L132/145: This section is outside my expertise and would need to be reviewed by an expert.
L152: This sentence is not correct- need to be re-written .
L183: “Red lines are the monitoring model of each year.” à sentence not needed.
Figure 5: can you use a color ramp with year so we can see any potential trend in the temperature seasonal patterns (e.g. recent years are in the reds and past years are in the oranges or blues). The caption is not correct, replace “Time series” by seasonal pattern. And explain what the blue dots and the red curves are clearly.
L189/231: this section need to be reviewed by an expert.
Results:
L253/254: The sign of deformation need to be explained much earlier in the paper and included in the abstract in parenthesis.
L259: “based on previous investigations” include citations.
L267/270: This belongs to the method section.
L271: when was the QTR built?
Figure 8: the range of the y-axis should be the same in both panels. Actually – the two curves should be overlay in the same graph to improve interpretation. Please include the location of the three regions of analysis used in the next section (Beiluhe etc…).
L274: The construction year should be provided much earlier in the manuscript.
L275: Why were the ERS-1 data not analyzed ? It would be important to assess the effect of QTR construction on permafrost deformation!
L276: On what base these four regions were defined? how are they different from the three regions detailed in the next section.
L297: what LOS means – again, you need to use homogenous terminology
Fig 9,10: The location where the pictures in figure 10 have been taken should be included in figure 9.
L337/340: This should not be in the main text but in the caption of the figure.
L354/359: This belongs to the discussion.
Discussion
L368/376: This belongs to the method section!!!
L377/384: This belongs to the result section!!!
Author Response
The manuscript in its current state is not acceptable for publication at Sensors. Following are the main issues that would need to be addressed for the study to be considered for publication:
Extensive revisions of the English expression is required – the text is barely understandable in several places. Examples are provided for the abstract in the minor comments section below to show the extent of the revision required.
Response: thanks, we have improved the English expression based on your suggestions in the new version. The whole manuscript has undergone language editing by experienced, native English speaking editors.
The title is too long, too technical.
Response: thanks, the title has been adjusted in the new version. (please see lines 2-4)
The text needs to be re-organized in several places as methodological details are now in the result and even the discussion section and discussion elements are in the result section (see comments below). The discussion is more or less an extension of the result section.
Response: thanks for your careful review and helpful suggestions.
The texts needed to be re-organized have been changed in the new versions. (please see lines 432-433, 442-448, 461-463)
The analysis could have been deepened much further without too much efforts: a better analysis of the spatio-temporal patterns of permafrost deformation would have been very valuable. For instance, a more straight forward assessment of the impact of the QTR construction on permafrost deformation could have been done by comparing in a more quantitative way deformation rates over time. Furthermore, similarly to the averaged deformation rates (over time) presented in figure 8, time series of averaged deformation rates (across space) would have been very nice to see, in relation to ground or air temperature time series.
Response: thanks for your suggestions.
In Figure 8, the deformation of the QTEC before the QTR construction has been given. Comparing the ground deformation before and after the QTR construction, it is easy to assessment of the impact of the QTR construction on permafrost deformation.
The terminology needs to be homogenized (e.g. permafrost motion / deformation rate).
Response: Thanks for your careful review.
The terminologies have bene homogenized in the new version. (please see line 161)
Some methodological details are missing (e.g. the reasoning behind the definitions of various regions for the analysis).
Response: Thanks, those details have been given in the new version.
Too many acronyms are used, making the reading difficult.
Response: thanks for your careful review.
Labeling of locations in the map figures need to be bigger.
Response: thanks, they have been changed in the new version.
MINOR COMMENTS
Abstract:
L15: what do you mean by “the embankment of the permafrost engineering”.
Response: I mean the roadbed of permafrost engineering, such as Qinghai-Tibet railway and Qinghai-Tibet highway.
L16: “the freezing and thawing effects of active layer” à “the freezing and thawing seasonal cycles”.
Response: thanks, it has been changed. (please see line 17)
L17: “reveal” à assess
Response: thanks, it has been changed. (please see line 19)
L18: “motions” à deformation
Response: thanks, it has been changed. (please see line 19)
L18: “from Wudaoliang to Tuotuohe section” à useless detail.
Response: thanks, it is removed. (please see line 19)
L19: are these “linear subsidence trends” based on interannual variations? If so, then change it to “interannual linear subsidence trends”.
Response: thanks, the linear subsidence trends is based on the interannual variations. ‘interannual’ has been added in the new version. (please see line 21)
L20: “introduced” à presented
Response: thanks, it has been changed. (please see line 22)
L20/21: “Experimental results show that the ground deformation rate along QTEC ranges from -5 to +5 mm/year during 1997-1999 observation period derived from ERS-1 data.” à Analysis of the ERS-1 data show ground deformation along QTEC ranging from -5 to +5 mm/year during the 1997-1999 observation period.
Response: thanks, it has been changed. (please see lines 22-25)
L22: “velocity” à rate. Be consistent in the terms you use.
Response: thanks, it has been changed.
L22: “primarily” à no needed
Response: thanks, it is removed. (please see line 26)
L24: reformulation needed, e.g. “Significant ground deformation was detected in three sections of the corridor using Sentinel-1A and InSAR imagery.”
Response: thanks, it has been changed. (please see lines 28-31)
L26/27: reformulation needed, e.g. “Ground deformation was associated with the development of thaw slumps in one of the three sections.”
Response: thanks, it has been changed. (please see lines 32-34)
Introduction:
L33: “is one of the most significant factors affecting” à “has the potential to affect”
Response: thanks, it has been changed. (please see line 41)
L64: If it is already known that InSAR is a promising technique, it doesn’t need to be the conclusion of your abstract.
Response: thanks, InSAR is a promising technique for monitoring ground deformation related to coal mining, landslide, earthquake. In the new version, the sentence has been removed.
Table 1 is not useful at this point. Furthermore, all the acronyms for the InSAR methods need to be explained in the caption.
Response: Thanks. Through Table 1, readers can know the recent applications of InSAR in QTP. So, we think it would be better to reserve the Table 1 in the introduction sections.
The acronyms for the InSAR methods have been explained in the sections.
L68: what does this expression “the relationship between permafrost and Qinghai-Tibet Plateau Engineering” mean? Do you mean the consequences of permafrost deformation on the Qinghai-Tibet Plateau Engineering development?
Response: I mean the consequences of Qinghai-Tibet Plateau Engineering development on permafrost deformation.
L77: PSI and SBAS: explain these acronyms.
Response: thanks for your careful review. These acronyms have been given in the new version. (see line 67)
Study area:
L90: “thermalkarst” à thermokarst
Response: thanks, it has been changed. (see line 113)
L90 and L95: references needed.
Response: thanks, references have been added. (see line 112 and 115)
Figure 1: explain SRTM and include a reference for the DEM map. The Amplitude is from which product?
Response: thanks. SRTM is short for shuttle radar topography mission. (please see line 120)
The amplitude map is from the Sentinle-1A images.
L93: mention the panels in Figure 2 associated with each land cover.
Response: thanks, we have added the others panel in Figure 2. (please see line 134)
L104: what does this means??? “Due to the constructions of those engineering, the statements of the permafrost have been broken.”
Response: thanks, I mean that the construction of QTR destroys the original hydrothermal balance of permafrost and leads to the degradation of permafrost. In the new version, we have changed this sentence. (please see lines 135-137)
L105: replace motion with deformation please.
Response: thanks, it has been changed.
Figure 3 no needed.
Response: the daily air temperature is used to calculate the ADDT and ADDF in this paper. So, I think that it would be better to add figure 3 in the paper.
L132: “all the SAR images are co-registrated” à what does this mean?
Response: sorry for misleading you.
The first step of InSAR is registration. The SAR images have to be precisely co-registered in order to extract the correct interferometric phase. The word has been corrected in the new version. (please see line 170)
L132/145: This section is outside my expertise and would need to be reviewed by an expert.
Response: Thanks. We have improved this section in the new version.
L152: This sentence is not correct- need to be re-written.
Response: thanks, the sentence has been changed. (please see line 193)
L183: “Red lines are the monitoring model of each year.” à sentence not needed.
Response: thanks, it has been removed. (please see line 231)
Figure 5: can you use a color ramp with year so we can see any potential trend in the temperature seasonal patterns (e.g. recent years are in the reds and past years are in the oranges or blues). The caption is not correct, replace “Time series” by seasonal pattern. And explain what the blue dots and the red curves are clearly.
Response: thanks. The caption has been changed.
The onsets of thawing and freezing season had been calculated (Table 1) and there is no potential trend. So, we do not use a color ramp with year. The onsets intervals of thawing and freezing season from 1997-2018 were marked in the figure.
Table 1 Onsets of thawing and freezing season from 1997-2018
Year |
Onset of thawing season (month-day) |
Onset of freezing season (month-day) |
R2 |
1997 |
5-10 |
9-26 |
0.91 |
1998 |
5-3 |
10-9 |
0.91 |
1999 |
4-21 |
9-19 |
0.82 |
2000 |
5-8 |
10-7 |
0.94 |
2001 |
5-10 |
10-11 |
0.91 |
2002 |
5-3 |
10-7 |
0.91 |
2003 |
5-5 |
10-11 |
0.9 |
2004 |
5-1 |
10-4 |
0.89 |
2005 |
5-5 |
10-8 |
0.91 |
2006 |
5-7 |
10-7 |
0.88 |
2007 |
5-4 |
10-10 |
0.87 |
2008 |
5-8 |
10-5 |
0.91 |
2009 |
4-30 |
10-11 |
0.89 |
2010 |
4-27 |
10-10 |
0.9 |
2011 |
5-6 |
10-11 |
0.91 |
2012 |
5-4 |
10-7 |
0.92 |
2013 |
4-29 |
10-4 |
0.89 |
2014 |
5-4 |
10-8 |
0.91 |
2015 |
4-29 |
10-12 |
0.9 |
2016 |
5-1 |
10-15 |
0.92 |
2017 |
5-4 |
10-15 |
0.87 |
2018 |
4-30 |
10-8 |
0.9 |
L189/231: this section need to be reviewed by an expert.
Response: Thanks. We have improved this section in the new version.
Results:
L253/254: The sign of deformation need to be explained much earlier in the paper and included in the abstract in parenthesis.
Response: The sign of deformation has been explained earlier in the paper. (please see line 294-295)
L259: “based on previous investigations” include citations.
Response: thanks, the references have been added. (please see line 353)
L267/270: This belongs to the method section.
Response: Thanks for your careful review. They have been moved to the study area section. (please see line 122)
L271: when was the QTR built?
Response: The QTR from Golmud to lhasa began to built in 2001 and completed in 2006.
Figure 8: the range of the y-axis should be the same in both panels. Actually – the two curves should be overlay in the same graph to improve interpretation. Please include the location of the three regions of analysis used in the next section (Beiluhe etc…).
Response: Figure 8 has been changed in the new version. (please see line 347)
L274: The construction year should be provided much earlier in the manuscript.
Response: thanks, the construction year have been given in the study area section. (please see line 110)
L275: Why were the ERS-1 data not analyzed? It would be important to assess the effect of QTR construction on permafrost deformation!
Response: thanks for your suggestions. We have added the ERS-1 results in the new versions. (please see line 347)
L276: On what base these four regions were defined? how are they different from the three regions detailed in the next section.
Response: thanks.
These four regions showed an obvious subsidence area, with the largest deformation rate being 17 mm/year from 2015-2018. So, we marked these four regions.
Regions A, B, and C are the three regions detailed in the nest section.
L297: what LOS means – again, you need to use homogenous terminology
Response: the LOS means light of sight.
Fig 9,10: The location where the pictures in figure 10 have been taken should be included in figure 9.
Response: thanks, the locations of Figure 10 have been given in Figure 9 in the new version. (please see line 385)
L337/340: This should not be in the main text but in the caption of the figure.
Response: thanks, we have changed the main text in the new version. (please see line 432)
L354/359: This belongs to the discussion.
Response: thanks, it has been moved to the discussion section. (please see lines 442-448)
Discussion
L368/376: This belongs to the method section!!!
Response: thanks for your comments. we have deleted the equation (5) in the new version. Reader could find the equation in the reference. (please see line 461)
L377/384: This belongs to the result section!!!
Response: thanks.
The object of this paper is to retrieve the ground deformation along QTEC using time-series InSAR. In section 5.1, we want to analyze the relationship between ground deformation and permafrost thermal statement. So, we calculate the mean annual ground temperature and describe the result in this section.

Round 2
Reviewer 2 Report
The revised manuscript has improved its clarity, but it seems not unacceptable to me that certain key parts of an InSAR study are still missing:
(a) How did the authors determine the length of baseline?
(b) Atmospheric corrections for the troposphere (vapor part) and ionosphere impacts are not presented.
(c) For the comment "it is unclear if the deformation data are consistent among different sensor records; and if sensor-related biases exist", no direct quantification is provided.
Other issues:
Line 470 "temperature baseline": such errors must be avoided!
Author Response
The revised manuscript has improved its clarity, but it seems not unacceptable to me that certain key parts of an InSAR study are still missing:
(a) How did the authors determine the length of baseline?
Response: Thanks for your comments.
In this paper, the small baseline strategy was applied to suppress the temporal decorrelation. Different small baseline threshold values were used for the different SAR stacks. Through previous studies, the temporal decorrelation is serious in permafrost region, so the temporal baseline (350 days) is no longer than one year. Consideration the orbit accuracy of different sensors and the time sampling of SAR images, the normal baseline threshold values are 800m, 500m, and 200m for ERS, ENVISAT and Sentinel-1A, respectively. After that, all the generate interferograms were checked, and the interferograms with serious decorrelations were rejected for deformation retrieval. (please see lines 156-159)
(b) Atmospheric corrections for the troposphere (vapor part) and ionosphere impacts are not presented.
Response: in the new version, the atmospheric corrections has been presented.
The residual phases for each interferogram were calculated by subtracting the estimated LP deformation and topographic error phase from the differential interferograms and unwrapped by the sparse Minimum Cost Flow (MCF) method (Costantini, 1998). The atmospheric phase was considered to consist of two components: topography related and non-topographic related (Ge et al., 2014). The two components were estimated separately. The topography related component can be estimated by the M-estimated. The non-topographic related atmospheric phase component is highly correlated in space but poorly in time, which can be estimated from the resultant phase based on the low pass filtering operation in spatial domain and high pass filtering operation in the temporal domain. After removing the APS from each interferogram and applying additional least-square estimation, we obtained the time-series deformation map. (please see lines 237-247)
Reference:
Ge, L., Ng, A. H. M., Li, X., Abidin, H. Z., & Gumilar, I. (2014). Land subsidence characteristics of Bandung Basin as revealed by ENVISAT ASAR and ALOS PALSAR interferometry. Remote Sensing of Environment, 154, 46-60.
Costantini, M. (1998). "A novel phase unwrapping method based on network programming." Geoscience and Remote Sensing, IEEE Transactions on 36(3): 813-821.
(c) For the comment "it is unclear if the deformation data are consistent among different sensor records; and if sensor-related biases exist", no direct quantification is provided.
Response: Thanks for your comments.
The deformation data are consistent among different sensor records. Different sensor has been jointly used to retrieve long-term ground deformation in several places (Hu et al., 2017; Liu et al., 2018; Ma et al., 2019). Hu et al., (2017) used the multi-spaceborne SAR images of ENVISAT, ALOS PALSAR-1, and Sentinel-1A to investigate the dynamics of consolidation settlement over the tailings impoundment area in the vicinity of Great Salt Lake, Utah from 2004-2016. Liu et al., (2018) monitored the land subsidence in Taiyuan from 1992 to 2015 using SAR images of ERS-1, ENVISAT, TerraSAR-X and Radarsat-2. Cross-validation has been carried out between descending-track Envisat ASAR and ascending-track TerraSAR-X images acquired from 2009 to2010, with minimum deformation rate differences of 1.7 mm/a. Ma et al., used Sentinel-1, COSMO-SkyMed and TerraSAR-X images to jointly reveal multi-scale subsidence of the Guangdong–Hong Kong–Macao Greater Bay Area. They found that the measurement from the COSMO-SkyMed and Sentinel-1 data both agree with the leveling data, with the deformation rate difference less than 3 mm/year. Through the above studies, we can see that deformation data are consistent among different sensor records if processing with the suitable InSAR method.
To direct qualify the results of the three sensors, the leveling measured should be collected or there is a common observation period between different SAR images. In the study, the leveling data haven’t been collected. The different SAR images don’t have a common observation period. So, it is difficult for us to directly quantify the consistency of the three results.
It is assuming that the topography changed little during the whole observation. We have compared the estimated DEM error of the results. Because there are very limited images for ERS-1 images, which may result in less accuracy result. So, the Sentinle-1A and EVNISAT results are compared to validate the estimated results. Figure 1 shows the difference between Sentinel-1A and ENVISAT DEM error along the QTR profile from P1 to P2. We can see that most of the DEM error difference in the ranges from -5 m to 5m, which could indirectly the consistency of the estimated DEM error from the two SAR stacks. The validity of the estimated DEM error can some extent validate the estimated deformation results.
Figure 1. the DEM error difference between Sentinel-1A and ENVISAT results (see pdf file)
Reference
Hu, X., Oommen, T., Lu, Z., Wang, T., & Kim, J. W. (2017). Consolidation settlement of Salt Lake County tailings impoundment revealed by time-series InSAR observations from multiple radar satellites. Remote sensing of environment, 202, 199-209.
Liu, Y., Zhao, C., Zhang, Q., Yang, C., & Zhang, J. (2018). Land subsidence in Taiyuan, China, monitored by insar technique with multisensor sar datasets from 1992 to 2015. IEEE Journal of Selected Topics in Applied Earth Observations and Remote Sensing, 11(5), 1509-1519.
Ma, P., Wang, W., Zhang, B., Wang, J., Shi, G., Huang, G., ... & Lin, H. (2019). Remotely sensing large-and small-scale ground subsidence: A case study of the Guangdong–Hong Kong–Macao Greater Bay Area of China. Remote Sensing of Environment, 232, 111282.
Other issues:
Line 470 "temperature baseline": such errors must be avoided!
Response: Thanks, it has been corrected in the new version. (please see line 482)

Reviewer 3 Report
The reviewers have addressed several of my comments successfully, but there are a few points that still need to be addressed before this manuscript can be considered for publication:
1- I still believe that the results shown in sections 5.1 and 5.2 have no place in the discussion section and should be moved to the result section. The discussion section is to discuss results presented in the result section, it is not a place to present new results.
2- I can see a few reference to "motion" in place of "deformation".
Author Response
The reviewers have addressed several of my comments successfully, but there are a few points that still need to be addressed before this manuscript can be considered for publication:
1- I still believe that the results shown in sections 5.1 and 5.2 have no place in the discussion section and should be moved to the result section. The discussion section is to discuss results presented in the result section, it is not a place to present new results.
Response: Thanks for your suggestion. Result and discussion have been reconstructed. Section 5.1 and 5.2 have been moved to the result section in the new version.
2- I can see a few reference to "motion" in place of "deformation".
Response: Thanks for your careful review, they have been replaced in the new version.

Round 3
Reviewer 2 Report
As a response to my previous comment "Atmospheric corrections for the troposphere (vapor part) and ionosphere impacts are not presented", the authors presented a paragraph describing the atmospheric correction methods but no relevant changes were shown in the following results and figures.
I wonder if the algorithm changes were actually made in the revision as what the authors described.
Author Response
As a response to my previous comment "Atmospheric corrections for the troposphere (vapor part) and ionosphere impacts are not presented", the authors presented a paragraph describing the atmospheric correction methods but no relevant changes were shown in the following results and figures.
I wonder if the algorithm changes were actually made in the revision as what the authors described.
Response: Thanks. I am sorry for misleading you.
As we known, the atmospheric corrections are important for the deformation retrieval in permafrost region, which have been considered and made in our algorithm. In the original version of the manuscript, the steps of the atmospheric corrections are not presented. But the results of the original version of the manuscript by the time-series InSAR considering atmospheric corrections, just as the flowchart of the time-series InSAR approach in Figure 6. In the previous, we have added the description of the atmospheric corrections as you suggested. In our previous other applications (Zhang et al., 2015a, 2015b, 2018; Wang et al., 2018), the atmospheric corrections are always considered.
Reference:
Zhang, Z., Wang, C., Tang, Y., Fu, Q., & Zhang, H. (2015a). Subsidence monitoring in coal area using time-series InSAR combining persistent scatterers and distributed scatterers. International journal of applied earth observation and geoinformation, 39, 49-55.
Zhang, Z., Wang, C., Tang, Y., Zhang, H., & Fu, Q. (2015b). Analysis of ground subsidence at a coal-mining area in Huainan using time-series InSAR. International Journal of Remote Sensing, 36(23), 5790-5810.
Zhang, Z., Wang, C., Wang, M., Wang, Z., & Zhang, H. (2018). Surface deformation monitoring in Zhengzhou city from 2014 to 2016 using time-series insar. Remote Sensing, 10(11), 1731.
Wang, C., Zhang, Z., Zhang, H., Zhang, B., Tang, Y., & Wu, Q. (2018). Active Layer Thickness Retrieval of Qinghai–Tibet Permafrost Using the TerraSAR-X InSAR Technique. IEEE Journal of Selected Topics in Applied Earth Observations and Remote Sensing, 11(11), 4403-4413.

This manuscript is a resubmission of an earlier submission. The following is a list of the peer review reports and author responses from that submission.
Round 1
Reviewer 1 Report
General comments:
The author used a total of 157 interferograms from three SAR satellites, including ERS-1/2: 1997-1999, ENVISAT: 2004-2010, and Sentinle-1A: 2015-2018, to quantify the ground deformation along the Qinghai-Tibet Plateau Engineer Corridor (QTEC). It involves considerable works. The scientific meaning is obvious. In the context of climate warming occurred on the Tibetan Plateau, the permafrost-induced subsidence along the QTEC puts threats on the operations of highway and railway. Using InSAR technique is a promising means to monitor the potential ground deformation. In recent years, many similar works have been done as summarized in Table 1, thanks to the authors to have a good review on them. Comparing with those studies, I don’t find obvious novelty from this study with respect to methodology and results. The results for three periods are likely subject to considerable uncertainty and inconsistency because they are obtained from different SAR satellites and by different processing methods. There are no direct validations made in this study, which is also a flaw seriously weakening the convincingness of the results. Some interpretations are potentially problematic. The current results may be misleading, therefore I would like to recommend a resubmission after well considering those issues.
Main concerns:
The SAR sources and their processing methods used in this study are different; I am afraid that the results are subject to different degrees of uncertainty. The impacts can be more or less observed from the studies listed in Table 1 that present varying results for the same location in a similar period. Therefore, the authors cannot use the results without any evaluation in advance. The results from different satellites and by different processing methods should be kept on a same level of uncertainty, otherwise the mean rate of deformation throughout the entire period makes no sense. In the engineering corridor, the causes of ground deformation on the engineering facilities and natural landscapes are quite different. Actually even the deformation of railway has occurred, it will be quickly repaired to ensure a safe pass of train. The cracks in Figure 10a and 10c may not link to the natural freezing and thawing cycle but the facilities themselves. In other words, the 20cm width crack on the slope of embankment does not mean the nearby natural permafrost surface suffers similar displacement. My suggestion is to separately discuss and analyze the deformations observed in natural locations and engineering infrastructure. The authors attempted to establish an agreement between a one-year seasonal NDVI shift (2018) and a mean rate of deformation through several years (2015-2018), as an important point in Conclusions. It is mathematically impossible. How can one-year seasonal variation affect inter-annual mean rate, nor vice versa? No direct validations have been made. It really hurt the convincingness of the results. For example, Figure 7c indicates tiny settlements have been observed from 2004 to 2018 near the Beiluhe Bridge. It might not be always true. The Beiluhe river often floods in summer and ruins the bridge. The maintenance work heavily disturbed the local conditions. There are many sites available for deformation observation along the railway, which can be collected for validating the results. It was concluded the first period (1995-1999), i.e., before the railway in operation, has smallest deformation. If the multiple year trend of deformation is caused by the freezing-thawing process occurred in the active layer in response to climate change, as explained in this manuscript, the warming temperature should be most important contributing factor. As a fact, the years 1997-1999 (especially 1999) are one of hottest periods on the plateau on record. I don’t quite understand why the deformation in that period is smallest in past decades. As I mentioned in point 2, making a distinction between natural surface and engineering infrastructure may help to clarify.
Specific Comments:
L122, five different satellites should be three. L128-129, SRTM DEM has a vertical error of about 15m on average and the errors vary in space. How do you consider this potential effect to the results? L143, What’s the basis for defining the scopes of spatial baseline and temporal baseline? Please provide reference. L138-146: more details in processing InSAR should be provided. For example, what kind of adaptive filter method is used? Any reference? Figure 4: texts too small and quality is bad. L165, eq. 1: as the equation is modified upon the original, is there any evaluation to ensure it works as expected? I don’t see any evaluations and it is unclear what this equation is for. Is it used to compute the mean rate? L169-170: the air temperature on the plateau is highly controlled by elevations, so you cannot use ADDT and ADDF from Wudaoliang site for everywhere along the corridor. L180: what are the thresholds specified for this study? L229, “three independent InSAR datasets show good consistency”, what kind of consistency? They are quite different as shown in Figure 5. L245, L291, L299, L311: Lots of words like “stable condition” and “similar trend” appear in the manuscript and they lack quantitative descriptions. Under what scope it is called stable condition? Some so-called “similar trend” are actually not really similar. Too many such ambiguous appear in the results section. Figure7, the vertical axis has a unit of m/year, which is obviously wrong. Also is problematic for the mean rate for Figure a. How do you compute the mean rate? The maps in Figure 6 and many other figures miss necessary legends. Figure 7b, c, if the location is close to the bridge, the real surface displacement should not be as stable as in those figures. Severe disturbance has been made to the nearby places due to heavy bridge work. L280-282: such crack (20cm in width) on the embankment can mean nothing about the displacement occurred on the railway. The embankment has been maintained to meet the requirement of a displacement less than 5 cm/a in order to secure the pass-through of a train running at a speed of 100km/hr. The authors present the results on three regions but with no further analyses of causes. L345-350: The linkage between the seasonal NDVI changes and the mean rate of deformation is groundless. It is mathematically impossible. L375: No direct observations have been collected for validating the results. It is especially inadequate for this study in wake of unknown uncertainties induced by different satellites, processing methods, and the use of equation (1). Validation and Limitation: The authors compared the results from previous studies and from this study. The authors stated that the permafrost deformation rate was in good agreement with others results. But the results in previous studies differ considerably from each other (Table1). For example, the deformation rate obtained by Chen et al. (2012) in 2001.44-2010.12 is -20-20mm/year (Beiluhe), while it is -8-2mm/year (Beiluhe) by Li et al. (2015) in 2003.4-2010.7. By contrast, the rate is -20-0mm/year in this study. They are at least not in agreement as good as claimed in terms of the range. The most important thing is we cannot trust your results without any validations with direct measurements.
Author Response
Comments 1
General comments:
The author used a total of 157 interferograms from three SAR satellites, including ERS-1/2: 1997-1999, ENVISAT: 2004-2010, and Sentinle-1A: 2015-2018, to quantify the ground deformation along the Qinghai-Tibet Plateau Engineer Corridor (QTEC). It involves considerable works. The scientific meaning is obvious. In the context of climate warming occurred on the Tibetan Plateau, the permafrost-induced subsidence along the QTEC puts threats on the operations of highway and railway. Using InSAR technique is a promising means to monitor the potential ground deformation. In recent years, many similar works have been done as summarized in Table 1, thanks to the authors to have a good review on them. Comparing with those studies, I don’t find obvious novelty from this study with respect to methodology and results. The results for three periods are likely subject to considerable uncertainty and inconsistency because they are obtained from different SAR satellites and by different processing methods. There are no direct validations made in this study, which is also a flaw seriously weakening the convincingness of the results. Some interpretations are potentially problematic. The current results may be misleading, therefore I would like to recommend a resubmission after well considering those issues.
Response: Thanks very much for your careful review and helpful comments. I have revised this paper based on your comments and suggestions.
Main concerns:
The SAR sources and their processing methods used in this study are different; I am afraid that the results are subject to different degrees of uncertainty. The impacts can be more or less observed from the studies listed in Table 1 that present varying results for the same location in a similar period. Therefore, the authors cannot use the results without any evaluation in advance. The results from different satellites and by different processing methods should be kept on a same level of uncertainty, otherwise the mean rate of deformation throughout the entire period makes no sense.
Response: Thanks for your comments.
I am sorry for mislead you. The objective of this paper is to retrieve the surface deformation along QTEC from the Wudaoliang to Tuotuohe section over the past 20 years. So, we collected three SAR datasets to achieve this goal: nine ERS-1 SAR images acquired from October 1997 to December 1999; thirty-nine ENVISAT SAR images from November 2004 to July 2010; forty Sentinel-1A SAR images from April 2015 to December 2018. In order to selected more inteferograms with high coherence, different spatial baseline thresholds are identified based on difference SAR dataset. But these three SAR datasets are retrieve using the same InSAR method in this paper. The results from the different SAR images are kept on a same level of uncertainty. Different SAR satellites are often jointly used to retrieve long-term ground deformation (Liu et al., 2018; Hu et al., 2017, 2018).
Reference:
Liu, Y.; Zhao, C.; Zhang, Q.; Yang, C.; Zhang, J.. Land subsidence in taiyuan, china, monitored by insar technique with multisensor sar datasets from 1992 to 2015. IEEE J-STARS 2018, 1-11.
Hu, X., Oommen, T., Lu, Z., Wang, T., & Kim, J. W. (2017). Consolidation settlement of Salt Lake County tailings impoundment revealed by time-series InSAR observations from multiple radar satellites. Remote sensing of environment, 202, 199-209.
Hu, X., Lu, Z., Pierson, T. C., Kramer, R., & George, D. L. (2018). Combining InSAR and GPS to Determine Transient Movement and Thickness of a Seasonally Active Low‐Gradient Translational Landslide. Geophysical Research Letters, 45(3), 1453-1462.
In the engineering corridor, the causes of ground deformation on the engineering facilities and natural landscapes are quite different. Actually even the deformation of railway has occurred, it will be quickly repaired to ensure a safe pass of train. The cracks in Figure 10a and 10c may not link to the natural freezing and thawing cycle but the facilities themselves. In other words, the 20cm width crack on the slope of embankment does not mean the nearby natural permafrost surface suffers similar displacement. My suggestion is to separately discuss and analyze the deformations observed in natural locations and engineering infrastructure.
Response: Thanks for your suggestions.
Previous studies indicate that the cracks on the embankment of QTR is the results of the cumulative effect of permafrost thawing and freezing (Chen et al., 2012). The 20cm width crack on the slope of embankment could reflect the deformation caused by the permafrost to some extent.
In the medium resolution SAR image, the QTR is chartered as a linear feature and only two or three pixels can be detected. It is difficult to monitor the deformation feature of different parts of the QTR. So, in this paper, we do not separate the engineering facilities and natural landscapes. The objective of this paper is to retrieve the surface deformation along QTEC from the Wudaoliang to Tuotuohe section over the past 20 years. In our another paper, the deformation of the QTR have been analyzed using high resolution SAR images, which is underreview (TerraSAR-X ST mode image).
The authors attempted to establish an agreement between a one-year seasonal NDVI shift (2018) and a mean rate of deformation through several years (2015-2018), as an important point in Conclusions. It is mathematically impossible. How can one-year seasonal variation affect inter-annual mean rate, nor vice versa? No direct validations have been made. It really hurt the convincingness of the results. For example, Figure 7c indicates tiny settlements have been observed from 2004 to 2018 near the Beiluhe Bridge. It might not be always true. The Beiluhe river often floods in summer and ruins the bridge. The maintenance work heavily disturbed the local conditions. There are many sites available for deformation observation along the railway, which can be collected for validating the results. It was concluded the first period (1995-1999), i.e., before the railway in operation, has smallest deformation. If the multiple year trend of deformation is caused by the freezing-thawing process occurred in the active layer in response to climate change, as explained in this manuscript, the warming temperature should be most important contributing factor. As a fact, the years 1997-1999 (especially 1999) are one of hottest periods on the plateau on record. I don’t quite understand why the deformation in that period is smallest in past decades. As I mentioned in point 2, making a distinction between natural surface and engineering infrastructure may help to clarify.
Response: Thanks for your suggestions.
We have deleted the section “Displacement and vegetation cover” and added the profile of QTR in this section 5.1 (please see lines 442- 496)
Specific Comments:
L122, five different satellites should be three.
Response: thanks for your comments. It has been corrected. (please see line 131)
L128-129, SRTM DEM has a vertical error of about 15m on average and the errors vary in space. How do you consider this potential effect to the results?
Response: Thanks for your comments. The Shuttle Radar Topography Mission (SRTM) DEM data with a 30m (1 arc-second) grid was used to remove the topographic phase in this study. The resolution of SRTM has been given in the new version.
In the process of generating the differential interferograms, we firstly converted the SRTM DEM from geographic coordinate system to Radar coordinate system and resampling the SRTM DEM in the 1 m, the same resolution of the multi-interferograms. Then the resampled SRTM DEM is used to remove the topographic phase. Even though there will introduce DEM error in the differential interferograms, in the parameter estimation step, we have considered those DEM error in the equation (1).
(1) (please see pdf file)
where is the long-term linear deformation, is the seasonal undulation, is the residual topographic contribution due to the DEM error. This is the most common method to estimate the DEM error of the time-series InSAR technique (Ferretti et al., 2001; Mora et al., 2003; Hooper et al., 2004).
Reference:
Ferretti, A., Prati, C., and Rocca, F.. 2001. “Permanent scatterers in SAR interferometry.” IEEE Transactions on Geoscience and Remote Sensing 39(1): 8-20. doi: 10.1109/36.898661.
Mora, O., Mallorqui, J. J., and Broquetas, A.. 2003. “Linear and nonlinear terrain deformation maps from a reduced set of interferometric SAR images.” IEEE Transactions on Geoscience and Remote Sensing 41(10): 2243-2253. doi: 10.1109/TGRS.2003.814657.
Hooper, A., Zebker, H. A., Segall, P., and Kampes, B.. 2004. “A new method for measuring deformation on volcanoes and other natural terrains using InSAR persistent scatterers.” Geophysical Research Letters 31(23). doi: 10.1029/2004GL021737.
L143, What’s the basis for defining the scopes of spatial baseline and temporal baseline? Please provide reference.
Response: Thanks for your comments. In the generation of interferograms, small spatial and temporal baseline would contribute to high coherence interferometric pairs (Lanari et al., 2004). (please see line 152)
Reference:
Lanari, R., Mora, O., Manunta, M., Mallorquí, J. J., Berardino, P., & Sansosti, E. (2004). A small-baseline approach for investigating deformations on full-resolution differential SAR interferograms. IEEE Transactions on Geoscience and Remote Sensing, 42(7), 1377-1386.
L138-146: more details in processing InSAR should be provided. For example, what kind of adaptive filter method is used? Any reference? Figure 4: texts too small and quality is bad.
Response: Thanks for your comments. More details have been given in processing InSAR and Figure 4 have been replaced in the new version. (please see line 162)
L165, eq. 1: as the equation is modified upon the original, is there any evaluation to ensure it works as expected? I don’t see any evaluations and it is unclear what this equation is for. Is it used to compute the mean rate?
Response: Thanks.
The Stefan equation (Nelson et al., 1997) is widely used to estimate the thaw depth. Liu et al. (2012) introduced a seasonal subsidence model based on the relationship between the seasonal thaw subsidence and the square root of the accumulated degree days of thaw (ADDT) through the Stefan equation. This equation model is used to compute the mean rate and seasonal deformation, has been tested in several places on the QTP (Liu et al., 2012, 2014; Wang et al., 2017; Chen et al., 2018). In our previous work, we have used the equation to estimation the active layer thickness in Beiluhe region (Wang et al., 2018). This equation is a great choice the retrieve ground deformation in permafrost region.
Reference:
Chen, J., Liu, L., Zhang, T., Cao, B., & Lin, H. (2018). Using Persistent Scatterer Interferometry to Map and Quantify Permafrost Thaw Subsidence: A Case Study of Eboling Mountain on the Qinghai‐Tibet Plateau. Journal of Geophysical Research: Earth Surface, 123(10), 2663-2676.
Wang, C., Zhang, Z., Zhang, H., Zhang, B., Tang, Y., & Wu, Q. (2018). Active Layer Thickness Retrieval of Qinghai–Tibet Permafrost Using the TerraSAR-X InSAR Technique. IEEE Journal of Selected Topics in Applied Earth Observations and Remote Sensing, 11(11), 4403-4413.
Wang, C., Zhang, Z., Zhang, H., Wu, Q., Zhang, B., & Tang, Y. (2017). Seasonal deformation features on Qinghai-Tibet railway observed using time-series InSAR technique with high-resolution TerraSAR-X images. Remote sensing letters, 8(1), 1-10.
Liu, L., Schaefer, K., Zhang, T., & Wahr, J. (2012). Estimating 1992–2000 average active layer thickness on the Alaskan North Slope from remotely sensed surface subsidence. Journal of Geophysical Research: Earth Surface, 117(F1).
Liu, L., Jafarov, E. E., Schaefer, K. M., Jones, B. M., Zebker, H. A., Williams, C. A., ... & Zhang, T. (2014). InSAR detects increase in surface subsidence caused by an Arctic tundra fire. Geophysical research letters, 41(11), 3906-3913.
L169-170: the air temperature on the plateau is highly controlled by elevations, so you cannot use ADDT and ADDF from Wudaoliang site for everywhere along the corridor.
Response: Thanks for your comments.
You are right. The air temperature on the plateau is highly controlled by elevations. However, we cannot get the daily air temperature along the corridor from 1997 to 2019. So, it is assuming the air temperature along the corridor from wudaoliang to tuotuohe is the same as the elevation doesn’t change greatly. Many other studies use on measurement site air temperature as the whole study area air temperature (Liu et al., 2012, 2014; Zhao et al., 2014).
Reference:
Liu, L., Schaefer, K., Zhang, T., & Wahr, J. (2012). Estimating 1992–2000 average active layer thickness on the Alaskan North Slope from remotely sensed surface subsidence. Journal of Geophysical Research: Earth Surface, 117(F1).
Liu, L., Jafarov, E. E., Schaefer, K. M., Jones, B. M., Zebker, H. A., Williams, C. A., ... & Zhang, T. (2014). InSAR detects increase in surface subsidence caused by an Arctic tundra fire. Geophysical research letters, 41(11), 3906-3913.
Zhao, R., Li, Z. W., Feng, G. C., Wang, Q. J., & Hu, J. (2016). Monitoring surface deformation over permafrost with an improved SBAS-InSAR algorithm: With emphasis on climatic factors modeling. Remote Sensing of Environment, 184, 276-287.
L180: what are the thresholds specified for this study?
Response: Thanks for your comments. The thresholds specified for this study have been given in the new version. (please see line 222)
L229, “three independent InSAR datasets show good consistency”, what kind of consistency? They are quite different as shown in Figure 5.
Response: we have rewritten the sentence in the new version.
L245, L291, L299, L311: Lots of words like “stable condition” and “similar trend” appear in the manuscript and they lack quantitative descriptions. Under what scope it is called stable condition? Some so-called “similar trend” are actually not really similar. Too many such ambiguous appear in the results section.
Response: thanks. Those words have been replaced in the new version.
Figure7, the vertical axis has a unit of m/year, which is obviously wrong. Also is problematic for the mean rate for Figure a. How do you compute the mean rate? The maps in Figure 6 and many other figures miss necessary legends.
Response: Thanks for your comments. Those figures have been replaced in the new version. (please see line 343)
Figure 7b, c, if the location is close to the bridge, the real surface displacement should not be as stable as in those figures. Severe disturbance has been made to the nearby places due to heavy bridge work.
Response: Thanks, you are right. The location is in the north of the bridge. Obvious deformation is detected in this region.
L280-282: such crack (20cm in width) on the embankment can mean nothing about the displacement occurred on the railway. The embankment has been maintained to meet the requirement of a displacement less than 5 cm/a in order to secure the pass-through of a train running at a speed of 100km/hr. The authors present the results on three regions but with no further analyses of causes.
Response: It is difficult to collect the levelling data of the QTR in this area, our estimated deformation results cannot be verified by levelling measurement directly. So, we want to use the subsidence phenomena found in the study area (such crack (20cm in width) on the embankment) to verify our measurement indirectly.
The section of the QTR embankment is trapezoidal. Figure 1 shows the measured QTR embankment profile in Beiluhe region. The ballast of the embankment has been maintained to meet the requirement of a displacement less than 5 cm/a in order to secure the pass-through of a train running at a speed of 100km/hr. Generally, the deformation occurs on the both sides of subgrade shoulder. such crack (20cm in width) are the result of cumulative deformation.
Figure 1. The measured QTR embankment profile in beiluhe region. (please see pdf file)
L345-350: The linkage between the seasonal NDVI changes and the mean rate of deformation is groundless. It is mathematically impossible.
Response: Thanks for your comments. The discussion of the displacement and vegetation cover has been removed in the new version.
L375: No direct observations have been collected for validating the results. It is especially inadequate for this study in wake of unknown uncertainties induced by different satellites, processing methods, and the use of equation (1).
Validation and Limitation: The authors compared the results from previous studies and from this study. The authors stated that the permafrost deformation rate was in good agreement with others results. But the results in previous studies differ considerably from each other (Table1). For example, the deformation rate obtained by Chen et al. (2012) in 2001.44-2010.12 is -20-20mm/year (Beiluhe), while it is -8-2mm/year (Beiluhe) by Li et al. (2015) in 2003.4-2010.7. By contrast, the rate is -20-0mm/year in this study. They are at least not in agreement as good as claimed in terms of the range. The most important thing is we cannot trust your results without any validations with direct measurements.
Response: Thanks for your comment.
Firstly, through several field investigations many subsidence phenomena were found in the study areas (pictures of Beiluhe in Figure 10), which could verify our monitoring subsidence results indirectly. On the other hand, in the tuotuohe region the water region change has been obtained and analyzed, which is related to the thawing of the permafrost.
In our results, the deformation rate is -20 to +10 mm/year. Chen et al. (2013) retrieved the ground deformation along QTR in Beiluhe area using C- and L-band small SAR interferometry. The estimated surface motion rate along embankment ranges from -20 to +20 mm/year. Li et al., (2015) monitored the surface deformation in Beiluhe area using InSAR with ENVISAT images. The deformation velocity near the QTR embankment is larger than -10 mm/year. Similar, our previous studies in Beiluhe regions with TerraSAR-X ST mode images show the similar deformation trends, with the motion rate ranges from -20 to 0 mm/year (Wang et al., 2018). The small differences between this paper and the previous studies due to the following aspects: 1) different band SAR images and InSAR processing method are used, which contributed to this difference; 2) the observation periods are difference, which would be another factor. We note that despite that these case studies are conducted at different time periods, the gradual subsidence trends are all on the order of centimeters per year, similar with our reported subsidence trends. It should be noticed that most of the previous studies used the SAR images acquired before 2010. In this paper, the latest ground deformation along QTEC have been obtained.
The leveling measurement is the most effective means of verification. In our further work, we will collect the ground measurement data to validate our result directly.
Reference:
Chen, F.; Lin, H.; Zhou, W.; Hong, T.; Wang, G.. Surface deformation detected by ALOS PALSAR small baseline SAR interferometry over permafrost environment of Beiluhe section, Tibet Plateau, China. Remote Sens. Environ. 2013, 138, 10-18.
Li, Z.; Tang, P.; Zhou, J.; Tian, B.; Chen, Q.; Fu, S. Permafrost environment monitoring on the Qinghai-Tibet Plateau using time series ASAR images. Int. J. Digit. Earth 2015, 8(10), 840-860.
Wang, C.; Zhang, Z.; Zhang, H.; Zhang, B.; Tang, Y.; Wu, Q.. Active Layer Thickness Retrieval of Qinghai–Tibet Permafrost Using the TerraSAR-X InSAR Technique. IEEE J-STARS 2018, 11(11), 4403-4413.

Reviewer 2 Report
This manuscript presents an InSAR time-series study over the Qinghai-Tibet Engineering Corridor (QTEC) in Tibet, where the degradation of the permafrost affects infrastructures development. Several studies of InSAR monitoring of ground deformation have been already undertaken in this area (Table 1) but the authors here combined the measurements of 3 sensors to obtain a long period time-series from 1997 to 2018. Although the quantity of processed data has to be underlined and I believe that the mapping of the deformed areas and the quantification of the rates is very valuable for the engineering and scientific community, I think that the manuscript presents several issues as much in the InSAR processing than in the quantification of the uncertainties or in the interpretation of the results, which need to be addressed. Here are my major concerns about the manuscript:
1) The authors model the InSAR ground deformation data with a periodic deformation function proportional to the freezing and thawing degree days, plus a linear trend. Why did the authors choose to include this periodic function if the results of this periodic decomposition are never shown, analyzed or discussed in the manuscript? Freezing and thawing coefficients maps (At, Af) obtained from equation (1) could be shown in map view to visualize the lateral variability of the amplitudes of the freeze-thaw cycles. Also, the authors may only interpret rates for pixels with large periodic cycles to only include pixels affected with freeze/thaw cycles processes and not analyze ground deformation rates related to other processed. Also, this choice of periodic function proportional to the square root of the degree day, contrary to a sinusoidal basis function with variable amplitudes and timing of maximum amplitude, assumes that the surface temperatures and thawing/freezing onsets are uniform and fix within the study area. Are they? This may be discussed in the text.
On the contrary, if the authors decide to only study the long-term linear changes of ground displacements, as it is done currently in the text, then I do not see the need to include this additional periodic function that mainly only increases the variance and trade-offs of the model parameters of equation 3. In any case, time series models in Fig 7, 9,12 and 14 should be shown to visualize the agreement of the model with the data, because, unless I make a mistake, the authors do not show at any time the results of the model in the current version of the manuscript.
2) The authors used 3 different data sets (namely ERS, Envisat, and Sentinel-1), which present 3 distinct temporal sampling and geometrical baselines that improve with time (Fig. 4). For a robust comparison and the reliability of the results, some uncertainties should be included. More particularly, ground velocity obtained with ERS (with poor temporal sampling and large perpendicular baselines) should have higher uncertainties than the one obtained with Envisat or Sentinel-1. Time-series examples in Figs. 9, 12 and 14 clearly show that the rate obtained with ERS (which presents a variability of ~60mm) will largely depend on the reliability of the last acquisitions (which depends on the atmospheric conditions or the geometrical/perpendicular baseline) and will also depend on the fit to the periodic function (which is also poorly constrained by the lack of data for ERS). The manuscript may explain why the rates here differ with previous studies (Table 1) obtained on the same areas, and, therefore if they are consistent within the error bars. The authors review all the rates obtained in other studies in the Validation session (5.3) and stipule "consistency". However, given the large ranges of results in Table 1, I think that a more robust quantification of the uncertainties and difference is required. The validation section may include a scientific quantification of the errors rather than this brief comparison and a sentence about the complexity of the permafrost modeling and the difficulty linked with temporal decorrelation.
3) In their processing, the authors apply two major corrections :
- they first de-trend the data from a polynomial ramp (equation 2) and a phase/elevation term. Did the authors estimate the ramps on the whole frame or only on the data coverage as shown in the Fig. 1 in the bottom? I am concerned about the fact that the study area is only 5km-large with a majority of pixel affected by the permafrost related deformations. Therefore how reliable is this "flattening" processing step and how the author know they do not remove any deformation? I think that this ramp and phase/elevation estimation must be done on stable pixel only, i.e bedrock areas not affected by the seasonal F/T cycles and long-term rates. Also, the authors say l.190 that this correction is done on interferograms with obvious phase ramps. However, I also think that all corrections must be applied on all the interferograms to not introduce any signal within the time series and assure consistency in the reference frame within all interferograms. Please, provide more details on this processing step in the text and some correction examples showing the reliability of the procedure.
- they correct the data from an elevation error proportional to the geometrical baseline (equation 3). However, as shown in the Fig.4, there is a covariance between the temporal and the perpendicular baseline, which means that the deformation and the topographic terms in equation 3 are in trade-off. Therefore, I would like to see the results obtained for this DEM error in map view to verify that this additional elevation term did not modify the estimation of the deformation rates (more particularly for ERS and Envisat with large perpendicular baselines). This point meets with my first point about the variance/trade-off of the model parameters and the necessity to include a degree-day model if this one is not used and interpreted (more particularly for ERS time-series with very few acquisitions).
4) The authors compare the ground deformation rate maps to the vegetation index and stipule a relationship between both. Could this statement be quantified? What is the correlation coefficient between the deformation rate and the NDVI index? Why only comparing with the vegetation? Rouyet et al., 2019, “Seasonal dynamics of a permafrost landscape, Adventdalen, Svalbard, investigated by InSAR” and Daout et al., 2017, “Large scale InSAR monitoring of permafrost freeze-thaw cycles on the Tibetan Plateau, found a correlation between water availability and the magnitude and timing of the ground displacements. The authors here propose that the vegetation protects the permafrost from the solar radiation and therefore the thawing of the permafrost. However, we are here in a very dry and arid environment with very few and homogenize alpine meadow outside of the railway/highway, as shown in Fig. 15. Is this few vegetation really protecting the permafrost? Is there not any other correlations that could explain the distribution of the ground deformation rates that should be discussed in the text?
5) What are the reference points in the velocity maps of Fig.5 and the time series shown in Figs. 9, 12 and 14? In other words, those long-term rates are relative to what?
6) How does the author explains the uplift rates that reach 10 mm/yr? Does it correspond to an increase of permafrost/active layer? Or those 10mm/yr of uplift are within the uncertainties of the data and do not mean anything?
Minor comments:
Please add a sentence about the seasonal atmospheric noise is the time series data. What is the magnitude of the expected atmospheric delays in this area and how they compare with the deformation, more particularly the periodic freeze-thaw cycles? Photographies Fig.2D and Fig.10A are already in Wang et al., 2017, "Seasonal deformation features on Qinghai-Tibet railway observed using time-series InSAR technique with high-resolution TerraSAR-X images." Permafrost thawing should increase the active layer thickness and therefore the amplitude of the periodic cycles, as explained in Liu et al., 2015, “Remote sensing measurements of thermokarst subsidence using InSAR”. Did the authors found any changes in amplitudes with time? l 211: Did the authors took into account the variation of the incidence angle within the frame? (~20° for Sentinel) l. 159- 161. The authors state that they introduce a new deformation model. However, a degree-day model is commonly used in the permafrost literature and as been for example used for InSAR time series modeling in Liu et al., 2012, “Estimating 1992–2000 average active layer thickness on the Alaskan North Slope from remotely sensed surface subsidence” or in Daout et al., 2017, “Large scale InSAR monitoring of permafrost freeze-thaw cycles on the Tibetan Plateau”. l 371. "the change of water regions". what is it? how it is computed? l.76 A hybrid time-series methodology taking advantage of "THE merits" of PS and SBAS is used to "identifY" more measurement points. Please add a reference to this InSAR development, eg: Hooper et al., 2008, “A multi-temporal InSAR method incorporating both persistent scatterer and small baseline approaches”. I have spotted a lot of English and spelling mistakes. Here some examples but I think the authors may need to go through the whole paper carefully:
l.35. largest extenT
l 138. A multi-temporal InSAR data processing stratEgy to retrieve ground deformation IS used. Considering the different attribute of SAR stacks with different wavelengthS...
l 362 the same area THAN in Figure 13.
l 362 At least three areas, marked as R1, R2, and R3, have undergone thaw slumps during ...
l. 364 By comparing Fig 13 and 16, we can see...
From 2007 to 2018, the areas of thaw slumps, for regions noted R1, R2, and R3, have increased by 0.435, 0.679, 0.317 km2, respectively.
l 365. we can see that the distribution of thaw slumps area ARE consistent with the ground motion.
Author Response
Comments 2
This manuscript presents an InSAR time-series study over the Qinghai-Tibet Engineering Corridor (QTEC) in Tibet, where the degradation of the permafrost affects infrastructures development. Several studies of InSAR monitoring of ground deformation have been already undertaken in this area (Table 1) but the authors here combined the measurements of 3 sensors to obtain a long period time-series from 1997 to 2018. Although the quantity of processed data has to be underlined and I believe that the mapping of the deformed areas and the quantification of the rates is very valuable for the engineering and scientific community, I think that the manuscript presents several issues as much in the InSAR processing than in the quantification of the uncertainties or in the interpretation of the results, which need to be addressed. Here are my major concerns about the manuscript:
1) The authors model the InSAR ground deformation data with a periodic deformation function proportional to the freezing and thawing degree days, plus a linear trend. Why did the authors choose to include this periodic function if the results of this periodic decomposition are never shown, analyzed or discussed in the manuscript? Freezing and thawing coefficients maps (At, Af) obtained from equation (1) could be shown in map view to visualize the lateral variability of the amplitudes of the freeze-thaw cycles. Also, the authors may only interpret rates for pixels with large periodic cycles to only include pixels affected with freeze/thaw cycles processes and not analyze ground deformation rates related to other processed. Also, this choice of periodic function proportional to the square root of the degree day, contrary to a sinusoidal basis function with variable amplitudes and timing of maximum amplitude, assumes that the surface temperatures and thawing/freezing onsets are uniform and fix within the study area. Are they? This may be discussed in the text. On the contrary, if the authors decide to only study the long-term linear changes of ground displacements, as it is done currently in the text, then I do not see the need to include this additional periodic function that mainly only increases the variance and trade-offs of the model parameters of equation 3. In any case, time series models in Fig 7, 9,12 and 14 should be shown to visualize the agreement of the model with the data, because, unless I make a mistake, the authors do not show at any time the results of the model in the current version of the manuscript.
Response:
For why choose the deformation model.
The Stefan equation (Nelson et al., 1997) is widely used to estimate the thaw depth. Liu et al. (2012) introduced a seasonal subsidence model based on the relationship between the seasonal thaw subsidence and the square root of the accumulated degree days of thaw (ADDT) through the Stefan equation. This equation model is often used to compute the mean rate and seasonal deformation, has been tested in several places on the permafrost region (Liu et al., 2012, 2014) and QTP (Wang et al., 2017; Chen et al., 2018). In our previous work, we have used the equation to estimation the active layer thickness in Beiluhe region (Wang et al., 2018). This equation is a great choice the retrieve ground deformation in permafrost region. In this paper, we focus on only study the long-term linear changes of ground displacements. So, we do not show the seasonal amplitude of the study area.
Reference:
Nelson, F. E., Shiklomanov, N. I., Mueller, G. R., Hinkel, K. M., Walker, D. A., & Bockheim, J. G. (1997). Estimating active-layer thickness over a large region: Kuparuk River basin, Alaska, USA. Arctic and Alpine Research, 29(4), 367-378.
Chen, J., Liu, L., Zhang, T., Cao, B., & Lin, H. (2018). Using Persistent Scatterer Interferometry to Map and Quantify Permafrost Thaw Subsidence: A Case Study of Eboling Mountain on the Qinghai‐Tibet Plateau. Journal of Geophysical Research: Earth Surface, 123(10), 2663-2676.
Wang, C., Zhang, Z., Zhang, H., Zhang, B., Tang, Y., & Wu, Q. (2018). Active Layer Thickness Retrieval of Qinghai–Tibet Permafrost Using the TerraSAR-X InSAR Technique. IEEE Journal of Selected Topics in Applied Earth Observations and Remote Sensing, 11(11), 4403-4413.
Wang, C., Zhang, Z., Zhang, H., Wu, Q., Zhang, B., & Tang, Y. (2017). Seasonal deformation features on Qinghai-Tibet railway observed using time-series InSAR technique with high-resolution TerraSAR-X images. Remote sensing letters, 8(1), 1-10.
Liu, L., Schaefer, K., Zhang, T., & Wahr, J. (2012). Estimating 1992–2000 average active layer thickness on the Alaskan North Slope from remotely sensed surface subsidence. Journal of Geophysical Research: Earth Surface, 117(F1).
Liu, L., Jafarov, E. E., Schaefer, K. M., Jones, B. M., Zebker, H. A., Williams, C. A., ... & Zhang, T. (2014). InSAR detects increase in surface subsidence caused by an Arctic tundra fire. Geophysical research letters, 41(11), 3906-3913.
2) The authors used 3 different data sets (namely ERS, Envisat, and Sentinel-1), which present 3 distinct temporal sampling and geometrical baselines that improve with time (Fig. 4). For a robust comparison and the reliability of the results, some uncertainties should be included. More particularly, ground velocity obtained with ERS (with poor temporal sampling and large perpendicular baselines) should have higher uncertainties than the one obtained with Envisat or Sentinel-1. Time-series examples in Figs. 9, 12 and 14 clearly show that the rate obtained with ERS (which presents a variability of ~60mm) will largely depend on the reliability of the last acquisitions (which depends on the atmospheric conditions or the geometrical/perpendicular baseline) and will also depend on the fit to the periodic function (which is also poorly constrained by the lack of data for ERS). The manuscript may explain why the rates here differ with previous studies (Table 1) obtained on the same areas, and, therefore if they are consistent within the error bars. The authors review all the rates obtained in other studies in the Validation session (5.3) and stipule "consistency". However, given the large ranges of results in Table 1, I think that a more robust quantification of the uncertainties and difference is required. The validation section may include a scientific quantification of the errors rather than this brief comparison and a sentence about the complexity of the permafrost modeling and the difficulty linked with temporal decorrelation.
Response: Thanks for your comment.
Firstly, through several field investigations many subsidence phenomena were found in the study areas (pictures of Beiluhe in Figure 10), which could verify our monitoring subsidence results indirectly. On the other hand, in the Tuotuohe region the water region change has been obtained and analyzed, which is related to the thawing of the permafrost.
In our results, the deformation rate is -20 to +10 mm/year. Chen et al. (2013) retrieved the ground deformation along QTR in Beiluhe area using C- and L-band small SAR interferometry. The estimated surface motion rate along embankment ranges from -20 to +20 mm/year. Li et al., (2015) monitored the surface deformation in Beiluhe area using InSAR with ENVISAT images. The deformation velocity near the QTR embankment is larger than -10 mm/year. Similar, our previous studies in Beiluhe regions with TerraSAR-X ST mode images show the similar deformation trends, with the motion rate ranges from -20 to 0 mm/year (Wang et al., 2018). The small differences between this paper and the previous studies due to the following aspects: 1) different band SAR images and InSAR processing method are used, which contributed to this difference; 2) the observation periods are difference, which would be another factor. We note that despite that these case studies are conducted at different time periods, the gradual subsidence trends are all on the order of centimeters per year, similar with our reported subsidence trends. It should be noticed that most of the previous studies used the SAR images acquired before 2010. In this paper, the latest ground deformation along QTEC have been obtained.
The leveling measurement is the most effective means of verification. In our further work, we will collect the ground measurement data to validate our result directly.
Reference:
Chen, F.; Lin, H.; Zhou, W.; Hong, T.; Wang, G.. Surface deformation detected by ALOS PALSAR small baseline SAR interferometry over permafrost environment of Beiluhe section, Tibet Plateau, China. Remote Sens. Environ. 2013, 138, 10-18.
Li, Z.; Tang, P.; Zhou, J.; Tian, B.; Chen, Q.; Fu, S. Permafrost environment monitoring on the Qinghai-Tibet Plateau using time series ASAR images. Int. J. Digit. Earth 2015, 8(10), 840-860.
Wang, C.; Zhang, Z.; Zhang, H.; Zhang, B.; Tang, Y.; Wu, Q.. Active Layer Thickness Retrieval of Qinghai–Tibet Permafrost Using the TerraSAR-X InSAR Technique. IEEE J-STARS 2018, 11(11), 4403-4413.
3) In their processing, the authors apply two major corrections :
- they first de-trend the data from a polynomial ramp (equation 2) and a phase/elevation term. Did the authors estimate the ramps on the whole frame or only on the data coverage as shown in the Fig. 1 in the bottom? I am concerned about the fact that the study area is only 5km-large with a majority of pixel affected by the permafrost related deformations. Therefore how reliable is this "flattening" processing step and how the author know they do not remove any deformation? I think that this ramp and phase/elevation estimation must be done on stable pixel only, i.e bedrock areas not affected by the seasonal F/T cycles and long-term rates. Also, the authors say l.190 that this correction is done on interferograms with obvious phase ramps. However, I also think that all corrections must be applied on all the interferograms to not introduce any signal within the time series and assure consistency in the reference frame within all interferograms. Please, provide more details on this processing step in the text and some correction examples showing the reliability of the procedure.
Response: Thanks for your comments. I estimated the ramps on the whole frame. Two typical interferograms with obvious orbit errors and topographic related phase errors are shown in Figure 1. It would produce significant artifacts in the final displacement result if those phase ramps are not removed. Figure 1 (b) and (f) indicate the corrected interferograms using the phase ramps correction model (2). Most of the phase ramps (orbit error, topographic related atmospheric error) have been removed and the deformation signal are retained and observed in the corrected interferograms. The phase ramps correction model has been used in many studies (Sun et al., 2015; Zhang et al., 2014; Zhang et al., 2018).
Figure 1. Examples of the interferograms before and after correction. (a): interferogram 20170225-20170414 before correction; (b): interferogram 20170225-20170414 after correction; (c): interferogram 20150413-20150507 before correction; (d): interferogram 20150413-20150507 after correction; (please see pdf file)
Reference:
Sun, Q.; Zhang, L.; Ding, X.L.; Hu, J.; Li, Z.W.; Zhu, J.J. Slope deformation prior to Zhouqu, China landslide from InSAR time series analysis. Remote Sens. Environ. 2015, 156, 45–57.
Zhang, L.; Ding, X.; Lu, Z.; Jung, H.S.; Hu, J.; Feng, G. A novel multitemporal InSAR model for joint estimation of deformation rates and orbital errors. IEEE Trans. Geosci. Remote Sens. 2014, 52, 3529–3540.
Zhang, Z. , Wang, C. , Wang, M. , Wang, Z. , & Zhang, H. . (2018). Surface deformation monitoring in zhengzhou city from 2014 to 2016 using time-series insar. Remote Sensing, 10(11).
- they correct the data from an elevation error proportional to the geometrical baseline (equation 3). However, as shown in the Fig.4, there is a covariance between the temporal and the perpendicular baseline, which means that the deformation and the topographic terms in equation 3 are in trade-off. Therefore, I would like to see the results obtained for this DEM error in map view to verify that this additional elevation term did not modify the estimation of the deformation rates (more particularly for ERS and Envisat with large perpendicular baselines). This point meets with my first point about the variance/trade-off of the model parameters and the necessity to include a degree-day model if this one is not used and interpreted (more particularly for ERS time-series with very few acquisitions).
Response: As you said, the deformation rate of the ERS-1 is not very accurate because less SAR datasets and heterogeneous spatial-temporal baseline would contribute to less accurate estimation result. The estimated DEM error is shown in Figure 2. The DEM error ranges from -20 to 10 m, which is consistent with the r elative accuracy of the SRTM DEM provided by NASA.
Figure 2. Estimated DEM error in (a)1997-1999, (b) 2004-2010, (c) 2015-2018 derived from the ERS-1, ENVISAT and Sentinel-1A, respectively. (please see pdf file)
4) The authors compare the ground deformation rate maps to the vegetation index and stipule a relationship between both. Could this statement be quantified? What is the correlation coefficient between the deformation rate and the NDVI index? Why only comparing with the vegetation? Rouyet et al., 2019, “Seasonal dynamics of a permafrost landscape, Adventdalen, Svalbard, investigated by InSAR” and Daout et al., 2017, “Large scale InSAR monitoring of permafrost freeze-thaw cycles on the Tibetan Plateau, found a correlation between water availability and the magnitude and timing of the ground displacements. The authors here propose that the vegetation protects the permafrost from the solar radiation and therefore the thawing of the permafrost. However, we are here in a very dry and arid environment with very few and homogenize alpine meadow outside of the railway/highway, as shown in Fig. 15. Is this few vegetation really protecting the permafrost? Is there not any other correlations that could explain the distribution of the ground deformation rates that should be discussed in the text?
Response: Thanks for your suggestions.
We have deleted the section “Displacement and vegetation cover” and added the profile of QTR in this section 5.1 (please see lines 448-502)
5) What are the reference points in the velocity maps of Fig.5 and the time series shown in Figs. 9, 12 and 14? In other words, those long-term rates are relative to what?
Response: Thanks for your comments. The reference point is selected at the railway bridge. In the new version, the reference points have been added in Fig 5. (please see line 291)
6) How does the author explains the uplift rates that reach 10 mm/yr? Does it correspond to an increase of permafrost/active layer? Or those 10mm/yr of uplift are within the uncertainties of the data and do not mean anything?
Response: Thanks for your comments. As you said, the uplift rate is within the uncertainties of the data and do not mean anything.
Minor comments:
Please add a sentence about the seasonal atmospheric noise is the time series data. What is the magnitude of the expected atmospheric delays in this area and how they compare with the deformation, more particularly the periodic freeze-thaw cycles?
Response: Thanks for your comments.
As we know, the atmospheric error is large in the permafrost regions. In this paper, we have considered the effect of atmospheric noise. We have decreased the effect of atmospheric delay in two aspects: 1) the topography related atmospheric delay are removed using the equation (2); 2) the selected CPs are connected based on the Delaunay triangulation network, which is beneficial to further remove atmospheric errors. After those two processing, we are assuming that most of the atmospheric error are removed. As for the magnitude of the expected atmospheric delays, we do not have done some work and it is out the scope of this paper. In the further, we would study the effects of the atmospheric delay.
Photographies Fig.2D and Fig.10A are already in Wang et al., 2017, "Seasonal deformation features on Qinghai-Tibet railway observed using time-series InSAR technique with high-resolution TerraSAR-X images." Permafrost thawing should increase the active layer thickness and therefore the amplitude of the periodic cycles, as explained in Liu et al., 2015, “Remote sensing measurements of thermokarst subsidence using InSAR”. Did the authors found any changes in amplitudes with time?
Response: We have changed Figure 2D and Figure 10A in the new version.
In the last two years, the filed investigation havn’t been conducted. Through our result derived from SAR datasets, we find obvious changes in some setions of QTEC. In our further work, we will collect the levelling measurement and conduct the filed investigation on QTP to validate our observations.
l 211: Did the authors took into account the variation of the incidence angle within the frame? (~20° for Sentinel)
Response: Thanks for your comments. In this paper, the variation of the incidence angle hasn’t been considered.
159- 161. The authors state that they introduce a new deformation model. However, a degree-day model is commonly used in the permafrost literature and as been for example used for InSAR time series modeling in Liu et al., 2012, “Estimating 1992–2000 average active layer thickness on the Alaskan North Slope from remotely sensed surface subsidence” or in Daout et al., 2017, “Large scale InSAR monitoring of permafrost freeze-thaw cycles on the Tibetan Plateau”.
Response: Thanks for your comments. We have changed the sentence in the new version.
l 371. "the change of water regions". what is it? how it is computed?
Response: Thanks for your comments. Figure 16 (f) shows the changes of water region between 20070726 and 20180807. The water regions reflect the thawing of permafrost. The water regions polygon is extracted by manual and the area of the water region is computed in ArcGIS software.
l.76 A hybrid time-series methodology taking advantage of "THE merits" of PS and SBAS is used to "identifY" more measurement points. Please add a reference to this InSAR development, eg: Hooper et al., 2008, “A multi-temporal InSAR method incorporating both persistent scatterer and small baseline approaches”. I have spotted a lot of English and spelling mistakes. Here some examples but I think the authors may need to go through the whole paper carefully:
Response: Thanks for your comments. The reference has been added in the new version. (please see line 80)
l.35. largest extenT
Response: it is corrected in the new version. (please see line 33)
l 138. A multi-temporal InSAR data processing stratEgy to retrieve ground deformation IS used. Considering the different attribute of SAR stacks with different wavelengthS...
Response: it is corrected in the new version. (please see line 151)
l 362 the same area THAN in Figure 13.
Response: it is corrected in the new version. (please see line 513)
l 362 At least three areas, marked as R1, R2, and R3, have undergone thaw slumps during ...
Response: it is corrected in the new version. (please see line 514)
364 By comparing Fig 13 and 16, we can see...
From 2007 to 2018, the areas of thaw slumps, for regions noted R1, R2, and R3, have increased by 0.435, 0.679, 0.317 km2, respectively.
Response: it is corrected in the new version. (please see line 515)
l 365. we can see that the distribution of thaw slumps area ARE consistent with the ground motion.
Response: it is corrected in the new version. (please see line 544)

Round 2
Reviewer 1 Report
General comments
The revision cannot satisfy my concerns. Only easy corrections have been made. I argued more validations should be made prior to simply using the models or equations. The authors repeatedly told me other guys used them just as this and no real attentions were paid to them. I argued one site air temperature records cannot fully represent a long corridor area, and the authors told me there are no data available and other guys assumed same and already got published.
For example, the cracks appearing on the embankment of the railway can be caused by many possible reasons. It can be related to engineering as well as natural deformation. In this SAR approach, there is no way to detect the deformations caused by engineering as limited by the resolution as in the authors’ reply. Therefore, the cracks on the embankment is not a reliable evidence to support the deformation caused by the freezing/thawing processes. The authors are generally expected to seek other strong evidences. But the authors did nothing to do with this concern. Actually it is easy to find many evidences on this area able to support the subsidence related to freezing/thawing processes – this is why I am positive to the scientific meaning of this study. I am afraid the authors have never been there in this study area.
Another example, while the authors acknowledged the high controlling role of elevations on the air temperature variation, they still assumed uniform air temperature along the railway corridor. There are many alternative ways to obtain the air temperature distribution on a fine resolution, although they are not as accurate as site observations. They can provide much better depiction than assuming spatially constant. It is very important, because the temperature variables used in the equations are of accumulative type.
I showed two examples here but it is not mean I satisfy the revisions in response to other major concerns.
I do not really believe the authors seriously investigated my questions. When I pointed out there might be some problems on the relationship between NDVI and deformation as well as the controversy between the climate change and deformation, the authors simply removed those texts without any explanation. The controversy can be caused by misinterpretation of results or wrong results. The authors should seriously inspect on the cause of controversy and make sure the results are correct.
Therefore, my opinion is reject until considerable revisions have been made.
Reviewer 2 Report
Thanks to the authors for the answers and the adaptations of the manuscript. However, some points have not been addressed and remain, which I list below.
I do not see any problems that the authors only focus on the long-term linear changes of ground displacements. However, they include a complex function in their time series analysis (the Stephan equation). My main concern is that this function requires to fix the freezing and thawing onsets, while we clearly see in time series Figs. 7, 8, 12 and 14 a spatial and inter-annual variability of thawing and freezing onsets. Could the authors, first, discuss this important point in the text and, secondly and more importantly, add in the Figs. 7, 8, 12 and 14, the inverted model from equation 1? It will allow to visualise and check how good is this model and if it does not bias the linear estimation, more particularly for ERS data, made of sporadic acquisitions.
My feeling is that this complex formulation is not necessary for ERS data as it only increases the posterior uncertainties by adding complexity, more parameters, unnecessarily in the inversion. A good way to test the impact of the Stephan term in the inversion would also be to perform an inversion without this periodic term and see how much it changes the linear term.
Thank you for adding Fig. 2 in your response but I think it should be added in the manuscript and also think that the trade-offs between deformation and DEM error term shoulb be discussed in the text. My concern is again regarding the ERS analysis. What is the point to add a DEM error term and a periodic function in the inversion when you have only 9 acquisitions? Parameters are clearly not constrained and if the author would computed the posterior uncertainties from the inversion, as I have previously suggested, they could maybe quantify that, for ERS and maybe Envisat, it is better to just include a linear term to get more robust results and to not have 10mm/yr of posterior uncertainties on the linear trend.
Minor comments:
l. 272. of the average incidence angle
l. 174 multi-looking is likely not the same for ERS/Envisat and Sentinel because of their opposite pixel ratio. Please check.
The paper is still missing some citations of InSAR studies that have advance our understanding of the permafrost distribution and response to climate forcing by comparing the seasonal amplitudes of the freeze/thaw cycles as well as the long-term trends to geomorphology, topography and geology:
- Rouyet et al., 2019, Seasonal dynamics of a permafrost landscape, Adventdalen, Svalbard, investigated by InSAR, Remote Sensing of the Environment.
- Daout et al., 2017, Large scale InSAR monitoring of permafrost freeze-thaw cycles on the Tibetan Plateau, GLR.